# Range extender mediates long-distance enhancer activity

Grace Bower[1], Ethan W. Hollingsworth[1,2], Sandra H. Jacinto[1], Joshua A. Alcantara[1], Benjamin Clock[1], Kaitlyn Cao[1], Mandy Liu[1], Adam Dziulko[3], Ana Alcaina-Caro[4], Qianlan Xu[5,6,7], Dorota Skowronska-Krawczyk[5,6,7], Javier Lopez-Rios[4,8], Diane E. Dickel[3,13], Anaïs F. Bardet[9], Len A. Pennacchio[3,10,11], Axel Visel[3,10,12] & Evgeny Z. Kvon[1✉]

Although most mammalian transcriptional enhancers regulate their cognate promoters over distances of tens of kilobases, some enhancers act over distances in the megabase range[1]. The sequence features that enable such long-distance enhancer–promoter interactions remain unclear. Here we used in vivo enhancer-replacement experiments at the mouse *Shh* locus to show that short- and medium-range limb enhancers cannot initiate gene expression at long-distance range. We identify a *cis*-acting element, range extender (REX), that confers long-distance regulatory activity and is located next to a long-range limb enhancer of *Sall1*. The REX element has no endogenous enhancer activity. However, addition of the REX to other short- and mid-range limb enhancers substantially increases their genomic interaction range. In the most extreme example observed, addition of REX increased the range of an enhancer by an order of magnitude from its native 73 kb to 848 kb. The REX element contains highly conserved [C/T]AATTA homeodomain motifs that are critical for its activity. These motifs are enriched in long-range limb enhancers genome-wide, including the ZRS (zone of polarizing activity (ZPA) regulatory sequence), a benchmark long-range limb enhancer of *Shh*[2]. The ZRS enhancer with mutated [C/T]AATTA motifs maintains limb activity at short range, but loses its long-range activity, resulting in severe limb reduction in knock-in mice. In summary, we identify a sequence signature associated with long-range enhancer–promoter interactions and describe a prototypical REX element that is necessary and sufficient to confer long-distance activation by remote enhancers.

Transcriptional enhancers are abundant *cis*-acting non-coding genomic elements that activate gene expression across different cell types of the organism in response to internal and external signals[3–7]. Enhancers often regulate their target promoters across long genomic distances[1,2,8–10]. Disruption of this long-range gene regulation often leads to diseases ranging from developmental anomalies to cancer[8,11–14].

A central question in gene regulation is how remote enhancers precisely relay regulatory information to their target promoters located hundreds of thousands or even millions of base pairs away, often without affecting the expression of intervening genes. Enhancers come into physical proximity to activate target promoters, but the mechanisms that mediate these functionally important interactions are only partially understood[1,15]. Mounting evidence indicates that higher-order three-dimensional (3D) chromatin organization and structural CTCF and cohesin proteins support enhancer–promoter (E–P) interactions

by restricting E–P communication to topologically associating domains (TADs)[16–20]. However, the global disruption of higher-order 3D chromatin organization and loop extrusion in CTCF- and cohesin-depleted cells only partially impairs E–P communication and gene expression[21–25]. Furthermore, developmental gene expression is also surprisingly robust to CTCF-binding-site deletions and structural perturbations affecting TADs[26–31]. Thus, which additional genetic factors establish and maintain long-range E–P communication during mammalian development remains poorly understood[1,13,15,32]. Identifying such factors is crucial for achieving an in-depth understanding of developmental processes and delineating disease mechanisms linked to disruption of long-range gene regulation.

Here we identify a unique sequence signature comprising [C/T] AATTA homeodomain (HD) motifs that mediates long-range E–P communication in developing limb buds. Deletion of these motifs selectively

[1]Department of Developmental and Cell Biology, University of California Irvine, Irvine, CA, USA. [2]Medical Scientist Training Program, University of California Irvine, Irvine, CA, USA. [3]Environmental Genomics and Systems Biology Division, Lawrence Berkeley National Laboratory, Berkeley, CA, USA. [4]Centro Andaluz de Biología del Desarrollo (CABD), CSIC-Universidad Pablo de Olavide-Junta de Andalucía, Seville, Spain. [5]Department of Physiology and Biophysics, School of Medicine, University of California, Irvine, CA, USA. [6]Department of Ophthalmology, School of Medicine, University of California Irvine, Irvine, CA, USA. [7]Center for Translational Vision Research, School of Medicine, University of California, Irvine, CA, USA. [8]School of Health Sciences, Universidad Loyola Andalucía, Seville, Spain. [9]Institut de Génétique et de Biologie Moléculaire et Cellulaire (IGBMC), Université de Strasbourg, CNRS UMR7104, INSERM U1258, Illkirch, France. [10]US Department of Energy Joint Genome Institute, Walnut Creek, CA, USA. [11]Comparative Biochemistry Program, University of California, Berkeley, CA, USA. [12]School of Natural Sciences, University of California, Merced, CA, USA. [13]Present address: Octant, Emeryville, CA, USA. ✉e-mail: ekvon@uci.edu

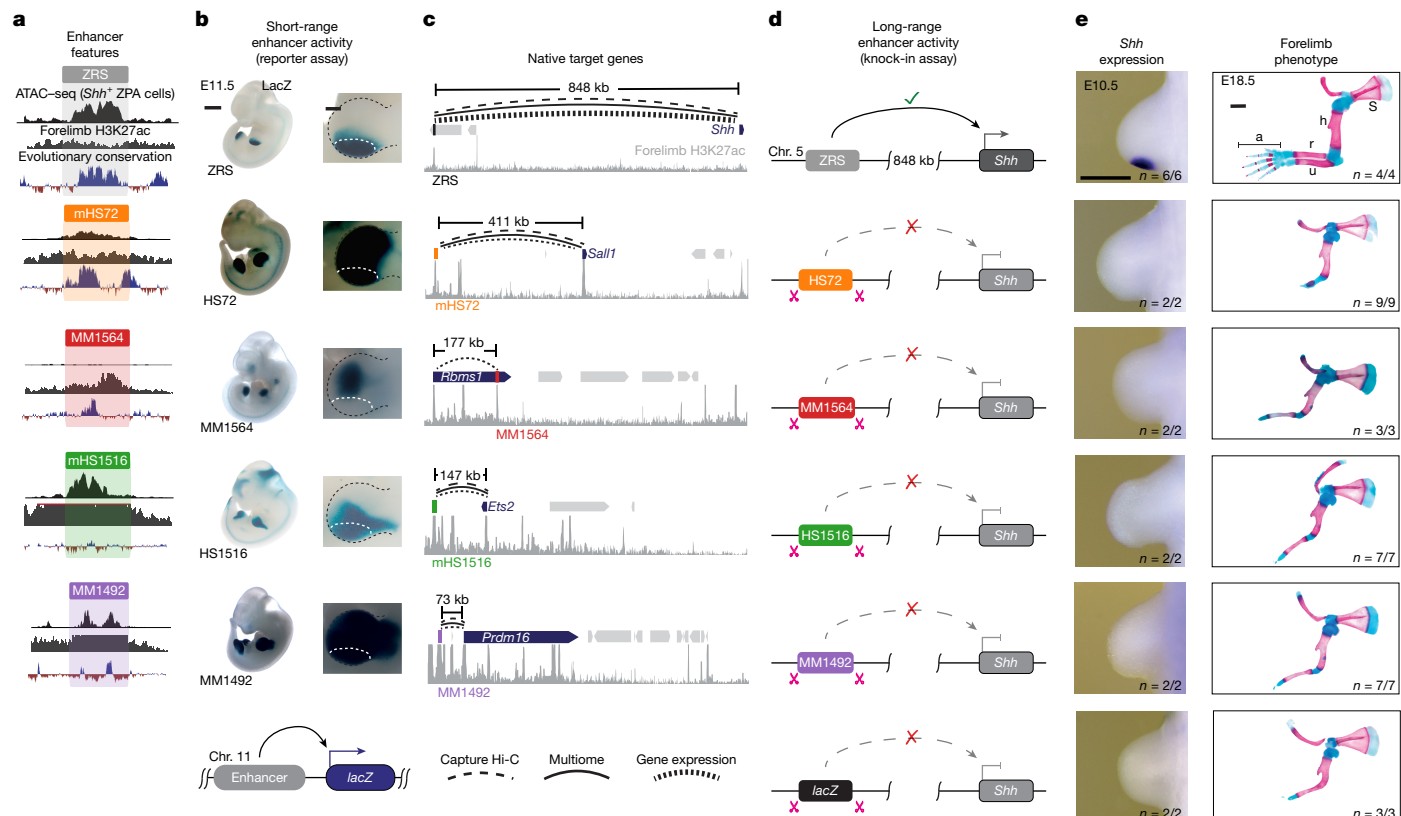

**Fig. 1 | Transplanted enhancers lack long-range limb activity in KI mice.** **a**, For each limb enhancer (coloured blocks), pseudobulk ZPA scATAC–seq, E11.5 forelimb H3K27ac chromatin immunoprecipitation–sequencing (ChIP–seq)[69] and placental conservation tracks are shown. mHS72 and mHS1516 are the mouse homologues of the HS72 and HS1516 enhancers. **b**, The corresponding enhancer activities in transgenic E11.5 mouse embryos. Images were acquired from the VISTA enhancer database[43]. Magnified forelimb buds are shown on the right. The white dotted lines demarcate the location of the ZPA. **c**, Mouse genomic map showing the location of corresponding enhancer regions and their endogenous target genes (dark blue). The curved lines indicate E–P links supported by capture Hi-C, multiome analysis of E11.5 mouse limb buds or matched enhancer activity and gene expression in limb buds of E11.5 or E10.5

embryos (Methods). The H3K27ac ChIP–seq signal from E11.5 forelimbs is shown underneath each region (grey). **d**, Schematic of mouse *Shh* loci with the ZRS enhancer replaced by the HS72, MM1564, HS1516 and MM1492 limb enhancers or a fragment of the *lacZ* sequence. **e**, Comparative *Shh* mRNA in situ hybridization analysis in WT and homozygous KI mouse embryos during forelimb bud development (first column). Corresponding skeletal preparations of E18.5 mouse embryos are shown in the second column. s, scapula; h, humerus; r, radius; u, ulna; a, autopod. The number of embryos that exhibited the representative limb phenotype over the total number of embryos with the genotype is indicated. Further details, including hindlimb analysis, are provided in Extended Data Fig. 2. Scale bars, 1 mm (**b** (left) and **e** (right)), 250 μm (**b**, right) and 500 μm (**e**, left).

abolishes distal enhancer activity but not its short-range ability. The [C/T]AATTA motifs are enriched at long-distance limb enhancer loci genome-wide and their presence correlates with E–P distance. We also characterize an extreme case in which the [C/T]AATTA motifs are located spatially separate from the enhancer within a genomic element, which we termed the REX element. The addition of the REX element can convert a short-range enhancer into a long-range enhancer that can act over 848 kb of genomic space. Our results indicate that short- or medium-range enhancers cannot function at long-distance range and establish that spatial specificity and long-distance activity are two separate and separable aspects of enhancer function.

## Medium-range enhancers cannot act at long range

To test the potential of short- and medium-range enhancers to act over long genomic distances, we performed a series of enhancer-replacement experiments at the *Shh* genomic locus. Limb-specific expression of *Shh* is controlled by the ZRS (also known as MFCS1), located at an extreme distance of about 848 kb from its target promoter in mice[2]. Mice deficient for the ZRS do not initiate any *Shh* expression in developing limb buds and display truncated limb phenotypes that are indistinguishable from those of mice with complete loss of SHH function in the limb[33]. Owing to the clear phenotypic readout and lack of redundancy, *ZRS-Shh*

is an ideal locus for assessing the long-range enhancer activity of transplanted enhancers.

To perform enhancer-replacement experiments, we selected four previously characterized developmental limb enhancers from other genomic loci on the basis of their ability to drive robust LacZ reporter expression in the developing limb bud mesenchyme (MM1492 and MM1564 (from mouse) and HS1516 and HS72 (from human); Fig. 1a,b). The HS72, HS1516 and MM1492 enhancers also drive LacZ reporter expression in the ZPA, a posterior domain of the limb mesenchyme where ZRS activates *Shh* gene expression (Fig. 1b). All four enhancers are evolutionarily conserved and marked by acetylation at lysine 27 in histone H3 (H3K27ac) in limb buds, a histone modification that is associated with active enhancers (Fig. 1a). To determine the genomic distance at which these enhancers act on their native promoters and to ensure that they are accessible to transcription factors (TFs) in the same cell types as the ZRS, we performed single-nucleus assay for transposase-accessible chromatin with sequencing (snATAC–seq) and single-nucleus RNA sequencing (snRNA-seq) multiome profiling of 14,000 cells from a wild-type (WT) embryonic day 11.5 (E11.5) hindlimb bud (Methods). After clustering all 14,000 cells on the basis of their chromatin and gene expression profiles, we annotated 16 clusters representing all major cell types in the developing limb bud (Methods, Extended Data Fig. 1 and Supplementary Table 1). The HS72, HS1516 and

MM1492 enhancers display strong DNA accessibility in *Shh* positive ZPA cells, consistent with their ability to drive robust reporter expression in posterior limb mesenchyme in transgenic mouse embryos (Fig. 1a).

To link limb enhancers to their putative target genes, we used the correlation between gene expression and open chromatin peaks from multiome profiling and E–P physical interaction data based on tissue-resolved high-resolution enhancer-capture Hi-C experiments (Methods). We identified *Prdm16* as the target for MM1492 (73 kb away), *Ets2* as the target for mHS1516 (mouse homologue of HS1516) (147 kb away) and *Sall1* as the target for mHS72 (mouse homologue of HS72) (411 kb away) (Fig. 1c). We then manually matched enhancer activity and the expression patterns of genes located within the same TADs in E11.5 mouse embryos. This analysis confirmed the above E–P links and additionally identified *Rbms1* as a putative target for MM1564 (177 kb away; Fig. 1b,c).

We next used genome editing to create a series of knock-in (KI) mice in which the functionally critical 1.3-kb region of the ZRS (Fig. 1d and Extended Data Fig. 2a,b) was replaced with selected limb bud enhancers from other genomic loci. Notably, replacing the mouse ZRS with HS72, MM1564, HS1516 or MM1492 limb enhancers resulted in a loss of detectable *Shh* expression in the limb bud (Fig. 1e). To assess the extent to which replacing the ZRS with a heterologous limb enhancer affects in vivo development, we examined the skeletal morphology in E18.5 KI mice. Consistent with the loss of limb-specific *Shh* expression, all four KI mouse strains displayed a truncated limb phenotype affecting both the forelimbs and hindlimbs. The limb phenotypes were indistinguishable from the phenotype caused by replacement of the ZRS with a nonfunctional control sequence of similar length (Fig. 1e and Extended Data Fig. 2c). These results indicate that, despite the presence of bona fide enhancer features capable of driving strong limb gene expression in the context of a transgene, all four enhancers lack long-range activity and cannot support *Shh*-mediated limb development.

## Transplanted enhancers are accessible

It is possible that silencing of the inserted region or failure of the insertions to create an open chromatin environment explains the inability of transplanted regions to act as long-range enhancers[34,35]. To test whether enhancers maintain their endogenous chromatin architecture at a remote ectopic location, we performed ATAC–seq experiments in ZPA region of forelimb and hindlimb buds of E11.5 mice heterozygous for the *ZRS^HS72* and *ZRS^HS1516* KI alleles.

HS72 and HS1516 are human enhancers that are highly conserved in mice but contain substitutions that allow discrimination of ATAC–seq reads from transplanted human and orthologous endogenous mouse enhancers (Methods and Extended Data Fig. 3). *ZRS^HS72* and *ZRS^HS1516* heterozygous mice formed normal limbs (Extended Data Fig. 2d), which enabled us to directly compare chromatin accessibility at the transplanted enhancer allele and the WT ZRS enhancer allele in fully developed limb bud tissue of the same mouse. Allele-specific ATAC–seq profiles demonstrated that both transplanted enhancers and orthologous mouse enhancers had an accessible open chromatin architecture (Fig. 2a). These results indicate that the heterologous HS72 and HS1516 limb enhancers were accessible when transplanted, but were unable to activate gene expression remotely, further supporting the idea that their inability to activate the *Shh* promoter in ZRS-replacement experiments is due to limitations in their genomic interaction range.

## The HS72 enhancer activates *Shh* at short range

The inability of heterologous limb enhancers to activate *Shh* expression was surprising, given their robust enhancer features and ability to establish open chromatin at the transplanted location. Previously, all four limb enhancers were characterized in transgenic mice using a well-established *hsp68* minimal promoter, but their compatibility with the *Shh* promoter

is unclear. To test whether E–P incompatibility[36,37] could have a role in the observed differences in gene activation, we placed the HS72 enhancer upstream of the mouse *Shh* promoter and the *LacZ* reporter gene and injected the resulting construct into fertilized mouse eggs. All transgenic mouse embryos (7 out of 7) displayed strong *LacZ* expression in the limb buds, which was identical to HS72's activity with the *hsp68* promoter, indicating that the HS72 enhancer is fully capable of activating the *Shh* promoter (Figs. 1a and 2b). Moreover, we previously showed that 32 enhancers characterised with the *hsp68* promoter maintained their in vivo activity when tested with the *Shh* promoter[38–40].

To test whether the HS72 enhancer can activate gene expression at the endogenous *Shh* locus, we generated a KI line in which the HS72 enhancer sequence was inserted around 2 kb upstream of the endogenous *Shh* promoter (Fig. 2c and Extended Data Fig. 4c,d). Preaxial polydactyly (formation of extra digits) provides a sensitive and readily accessible phenotypic readout of ectopic *Shh* misexpression beyond the ZPA of the developing limb[2,41]. As HS72 is active more broadly than the ZRS in limb mesenchyme, the presence of polydactyly would indicate that it can activate functional SHH signalling. Indeed, *Shh^HS72(+2kb)/WT* mice had polydactylous forelimbs and hindlimbs and de novo *Shh* expression in anterior hindlimb buds consistent with the HS72 enhancer driving expanded *Shh* activity[42] (Fig. 2c and Extended Data Fig. 4b). This result was similar to the polydactyly observed in transgenic mice in which the HS72::*Shh* transgene was integrated at a safe-harbour location in the mouse genome (Extended Data Fig. 4a). Importantly, *ZRS^HS72/WT* mice had normal limbs, indicating the absence of broad HS72-mediated *Shh* activation from a remote position (Extended Data Fig. 2d). Together, these results show that the HS72 enhancer can activate the *Shh* promoter at short range in its native endogenous context and in the context of a transgene.

## REX is required for long-range activation

The HS72 enhancer is compatible with the *Shh* promoter and is accessible when transplanted in place of the ZRS (Fig. 2) suggesting that its failure to activate *Shh* remotely could be due to the extreme distance from the *Shh* promoter. Indeed, the HS72, MM1564, HS1516 and MM1492 enhancers lie closer to their putative target genes than the ZRS (Fig. 1c). We hypothesized that there may be additional functional sequences associated with distant-acting enhancers that support long-range activation. To explore possible missing factors in long-distance activation, we examined the genomic region proximal to the HS72 enhancer, which has the longest native activity range among four heterologous enhancers at 411 kb away from the *Sall1* promoter (Fig. 1c). We identified a highly conserved block of sequence located adjacent to the HS72 enhancer (Fig. 3a). This sequence is not required to drive limb-specific activity in reporter assays (Figs. 1b and 2b and Extended Data Fig. 4), but, owing to its strong conservation and position, we hypothesized that this sequence might support enhancer activity, including the ability to activate gene expression over remote distances. To test this hypothesis, we generated KI mice in which the ZRS was replaced with an extended version of the HS72 enhancer that included this highly conserved upstream sequence (Fig. 3b).

Consistent with our prediction, addition of this uncharacterized sequence, which we termed the REX element, to the transplanted HS72 enhancer was sufficient to initiate *Shh* expression in the limb and resulted in full limb outgrowth with formation of all distal limb elements including fully formed zeugopod and autopod in *ZRS^HS72+REX/HS72+REX* mice (Fig. 3b and Extended Data Fig. 5a). *ZRS^HS72+REX/HS72+REX* mice also displayed polydactyly on both forelimbs and hindlimbs, consistent with ectopic *Shh* expression in limb buds driven by broader HS72 enhancer activity (Figs. 1b and 2b,c and Extended Data Fig. 5a–c).

Notably, the REX element alone does not act as a classical enhancer. When placed upstream of the minimal promoter in a transgene, it is unable to drive *LacZ* reporter expression in transgenic mouse

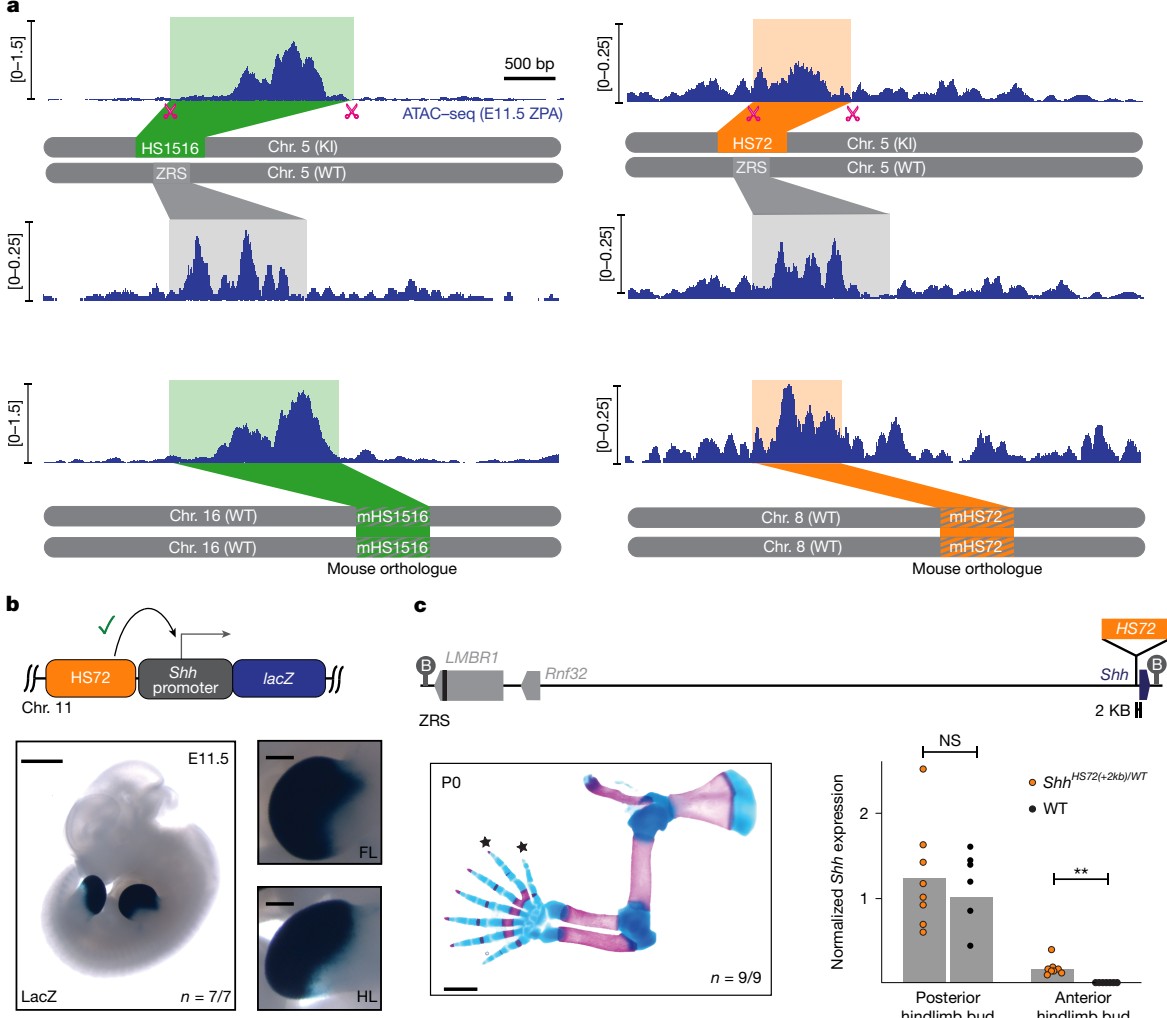

**Fig. 2 | Transplanted enhancers maintain open chromatin structure and drive functional *Shh* expression in the limb at short range. a**, Allele-specific ZPA ATAC–seq profiles at the transplanted HS1516 (green box) and HS72 (orange box) enhancers and the corresponding WT ZRS locus (top). Bottom, ATAC–seq profiles at endogenous mHS1516 (left; chromosome 16: 95847911–95849564; mm10) and mHS72 (right; chromosome 8: 89454508–89455383; mm10) enhancers (green and orange striped boxes). **b**, LacZ-stained transgenic E11.5 mouse embryo carrying the HS72 limb enhancer upstream of the *Shh* promoter (light blue) and *lacZ* reporter gene (dark blue). The number of embryos with robust LacZ staining in limb buds over the total number of transgenic embryos screened is indicated. **c**, Schematic of the $Shh^{HS72(+2kb)}$ KI allele, in which the HS72 enhancer was inserted around 2 kb upstream of the *Shh* TSS. Skeletal preparation of a forelimb from $Shh^{HS72(+2kb)/WT}$ mouse is shown below on the left. The star symbols indicate extra digits (polydactyly). Hindlimb images are shown in Extended Data Fig. 4b. Quantitative PCR (qPCR) analysis of gene expression in anterior and posterior E11.5 embryonic hindlimb buds in $Shh^{HS72(+2kb)/WT}$ ($n = 8$ embryos) and WT ($n = 6$ embryos) mice. Expression is normalized to the average *Shh* expression in the WT ZPA. Statistical analysis was performed using two-sided Wilcoxon rank-sum tests with no adjustments for multiple comparisons; **$P = 0.000311$. Scale bars, 1 mm (**b** (left) and **c**) and 250 μm (**b**, right).

embryos (Fig. 3c). Together, these results indicate that the REX element does not have enhancer activity on its own but is required for long-range heterologous enhancer activation at the *Shh* locus.

them at an ectopic remote location is sufficient to induce long-range gene activation. These results also suggest that the REX element can act in a modular manner to facilitate long-range enhancer action.

## REX is sufficient for long-range activation

To test whether the REX element can extend the range of heterologous short-range enhancers, we generated a KI mouse line in which the ZRS was replaced with a chimeric element consisting of the shortest-range heterologous limb enhancer from our test set (MM1492, 73 kb native E–P range) appended by the REX element (MM1492 + REX; Extended Data Fig. 5d). KI of this chimeric element was sufficient to induce *Shh* expression in the limb bud and resulted in a fully developed zeugopod and autopod in the $ZRS^{MM1492+REX/MM1492+REX}$ mice (Fig. 3d and Extended Data Fig. 5e). These mice also displayed polydactyly, consistent with broad limb activity of MM1492. Taken together, these experiments show that combining a short-range enhancer with the REX element and placing

## LHX motifs are critical for REX activity

To identify specific TFs that may be involved in REX-mediated long-range enhancer action, we examined potential TF-binding sites within the REX element. The REX element lacks CTCF and YY1 motifs and is not bound by CTCF in embryonic limb buds (Extended Data Fig. 6j). We scanned the REX element sequence for other candidate TF-binding sites and identified conserved motifs of which the sequences matched binding preferences for LHX2 and LHX9 homeodomain (HD) TFs as well as LEF1 (Fig. 4a and Extended Data Fig. 6a).

To determine the importance of LHX and LEF1 motifs for REX function, we generated two KI mice in which the ZRS was replaced with the extended HS72 sequence containing the REX element with either the

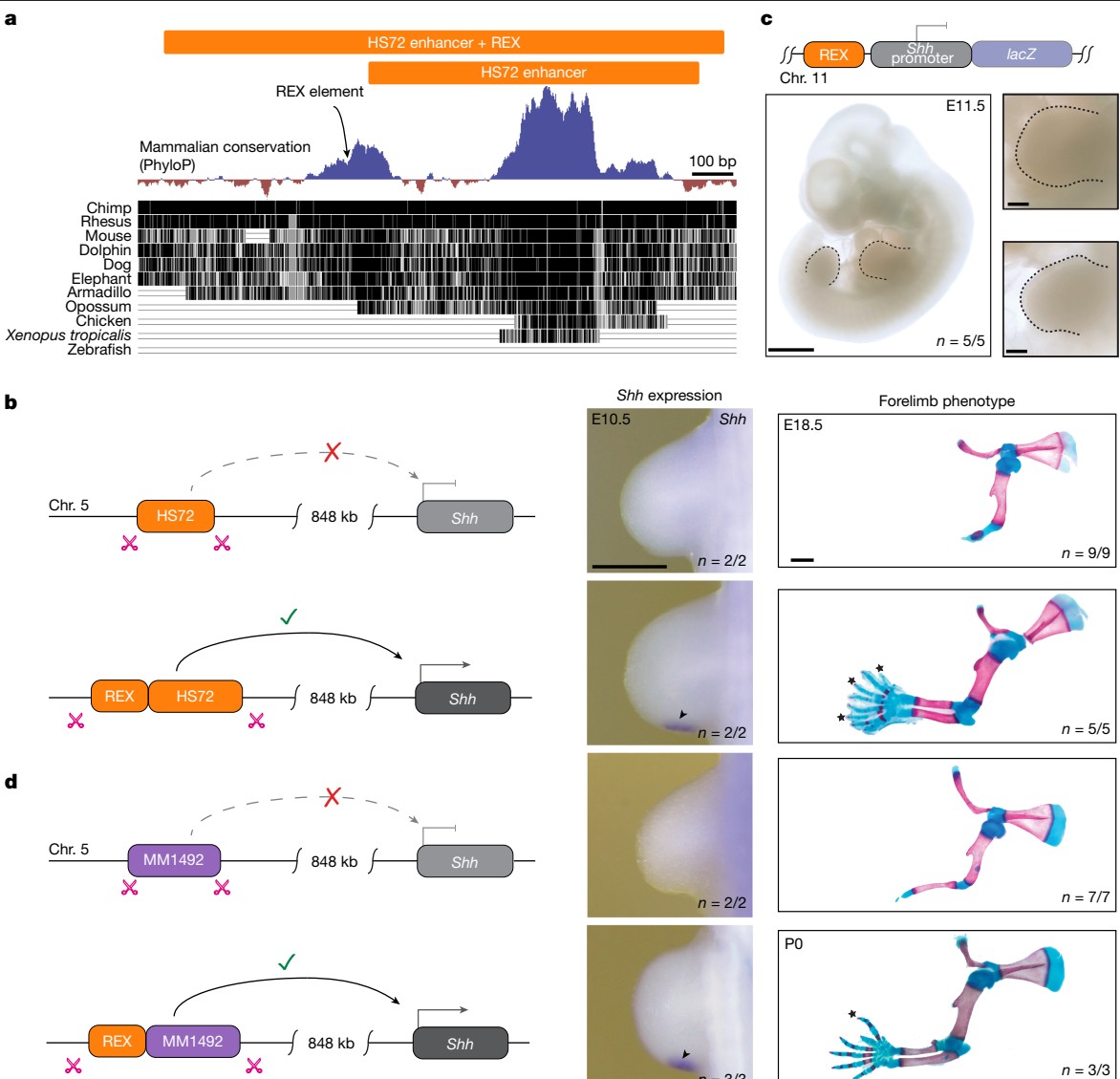

**Fig. 3 | The REX element is necessary and sufficient for long-range activation of *Shh* by a heterologous limb enhancer. a**, An evolutionary conserved element of unknown function is located adjacent to the human HS72 enhancer. The HS72 enhancer region is shown together with evolutionary conservation tracks. **b**, Replacement of the ZRS with an extended version of the HS72 enhancer (chromosome 16: 51623658–51625572; hg38) containing the REX element results in initiation of *Shh* expression in developing limb buds and full limb outgrowth in *ZRS^HS72+REX/HS72+REX* KI mice. Hindlimb and E11.5 *Shh* expression analysis is shown in Extended Data Fig. 5. **c**, The REX element lacks classical enhancer activity in E11.5 transgenic embryos when placed upstream of the *Shh* promoter and *lacZ* reporter gene. The number of transgenic embryos with no LacZ activity in the limb over the total number of transgenic embryos is indicated. **d**, Replacement of the ZRS with a chimeric *cis*-regulatory element consisting of the short-range MM1492 enhancer and the REX element from the HS72 enhancer region results in the initiation of *Shh* expression in developing limb buds full limb outgrowth in *ZRS^MM1492+REX/MM1492+REX* KI mice. The stars indicate extra digits (polydactyly). Hindlimb *Shh* expression analysis is provided in Extended Data Fig. 5. Scale bars, 1 mm (**c** (left) and **b** and **d** (right)), 500 µm (**b** and **d**, middle) and 250 µm (**c**, right).

single LEF1 motif or both LHX motifs mutagenized. Mice containing the REX element with mutated LEF1 motif showed fully developed limbs with polydactyly indicating that disruption of the LEF1 motif does not abolish long-range activity of the REX element (Fig. 4b). However, disrupting both LHX motifs resulted in mice with truncated limbs, indicative of a loss of *Shh* expression in the limb bud (Fig. 4c and Extended Data Fig. 6b). Together, these results demonstrate that the LHX motifs in the REX element are required for its ability to facilitate long-range enhancer activity at *Shh* locus.

## LHX motifs are enriched in remote enhancers

To investigate whether the LHX or other TF motifs are present at long-range limb enhancers, we used a collection of experimentally validated limb enhancers from the VISTA Enhancer Browser, a unique resource of human and mouse enhancers with in vivo activities characterized in transgenic mice[43]. We linked these bona fide limb enhancers to their putative target genes using the correlation between gene expression and open chromatin peaks from our single-cell ATAC–seq (scATAC–seq) and scRNA-seq experiment or E–P physical interactions from enhancer-capture Hi-C experiments (Extended Data Fig. 1d). We next performed motif analysis of these limb enhancers. We found 33 motifs that match the binding preferences of limb-expressed TFs that are significantly enriched in long-range acting (173 regions; E–P distance, 400 kb to 2 Mb) compared to short-range acting (28 regions; E–P distance, 10–200 kb) enhancer regions (false-discovery-rate-adjusted $P < 1 \times 10^{-2}$ and target frequency > 30%), with LHX2 and LHX9 being among the most significantly enriched (Fig. 4d, Extended Data Fig. 6d,e

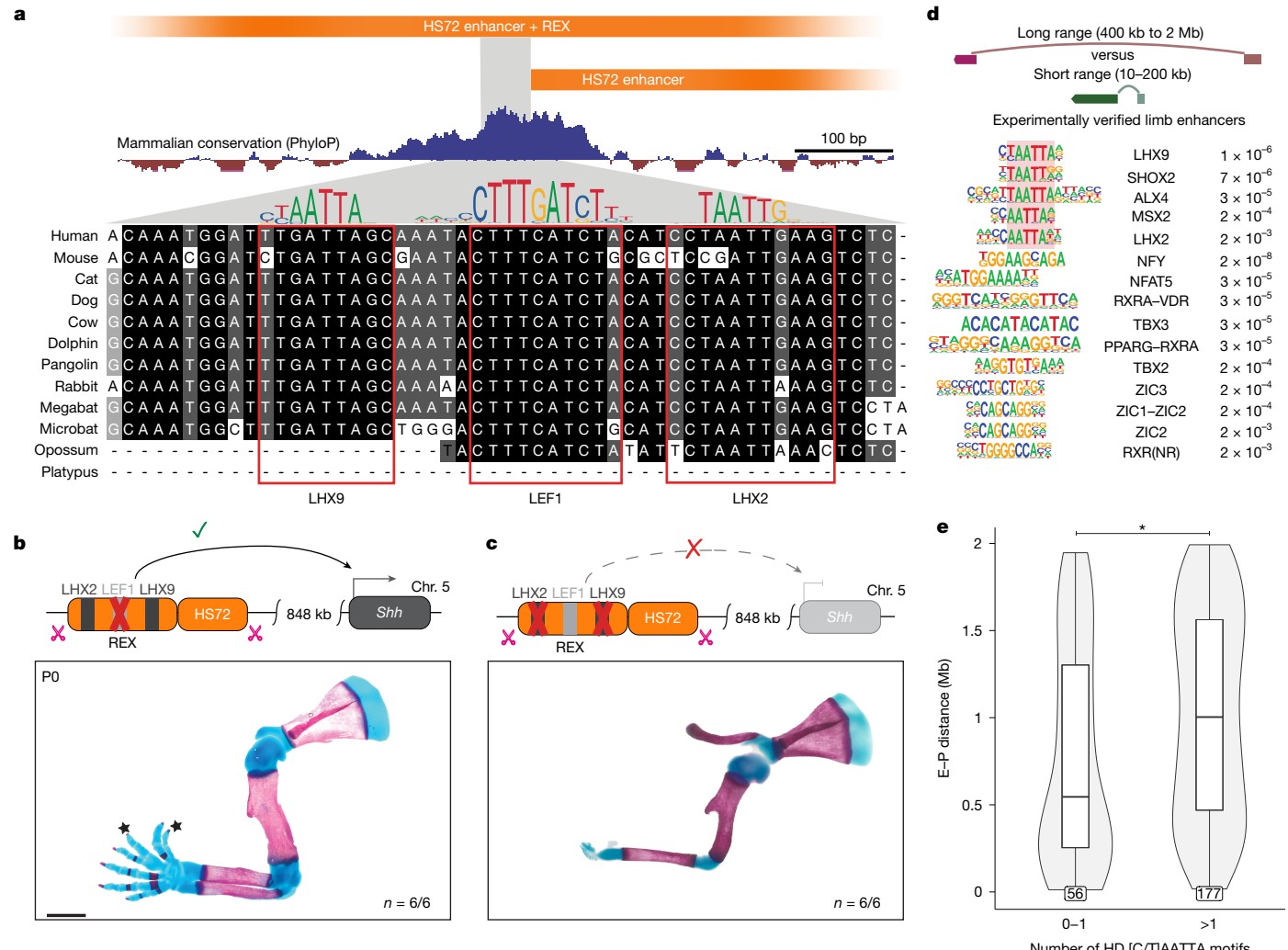

**Fig. 4 | The REX element contains conserved LHX motifs that are critical for its function and globally linked to long-range regulation. a**, The position and evolutionary conservation of predicted TF motifs within the REX element. The conserved core of the REX element (chromosome 16: 51624707–51624984; hg38) is aligned with orthologous sequences from 12 mammalian species. Sequences matching TF-binding preferences (below) are highlighted. **b,c**, Forelimb phenotype in $ZRS^{HS72+REX\Delta LEF1/HS72+REX\Delta LEF1}$ (**b**) and $ZRS^{HS72+REX\Delta LHX/HS72+REX\Delta LHX}$ (**c**) KI mice. The star symbols indicate extra digits (polydactyly). **d**, The top-most enriched TF motifs in long-range (400 kb to 2 Mb) compared with short-range (10 kb to 200 kb) enhancers for experimentally verified VISTA limb enhancers assigned to target genes by scATAC–seq/scRNA-seq or Capture Hi-C. Only the farthest target gene (within 2 Mb range) was considered for each enhancer. TF motifs are

grouped by their similarity. [C/T]AATTA HD motifs are highlighted by red shading. False-discovery rate values are shown on the right. Further details are provided in the Methods and Extended Data Fig. 6, and a complete list of motifs is provided in Supplementary Table 2. **e**, The distribution of E–P distances for VISTA limb enhancers assigned target genes by either Hi-C or Multiome ($n = 233$) with 0–1 or ≥2 conserved [C/T]AATTA HD motifs. Statistical analysis was performed using the two-sided Wilcoxon rank-sum test with no adjustment for multiple comparisons; $P = 0.0073$. The counts for each group are displayed in the outlined boxes at the base of the plot. The box plot shows the interquartile range (IQR; quartile 1 to quartile 3) (box limits), the median (centre line) and the minimum (Q1 − 1.5 × IQR) and maximum (Q3 + 1.5 × IQR) values (whiskers). Scale bars, 1 mm (**b** and **c**).

and Supplementary Table 2). Overall, HD TFs with a similar [C/T]AATTA consensus binding preference, including DLX5, MSX2 and SHOX2, comprised 10 out of these 33 motifs. By contrast, short-range limb enhancers showed no enrichment for TF motifs over long-range enhancers.

We next examined all 19,276 limb mesenchyme E–P pairs predicted from our scATAC–seq/scRNA-seq experiment (Extended Data Fig. 6c and Supplementary Table 2). Genome wide, the LHX2 motif was among the most significantly enriched in long-range versus short-range limb enhancers (CTAATTA, $P < 6 \times 10^{-32}$). Overall, [C/T]AATTA HD motifs, including both LHX2 and LHX9 motifs, comprised 15 out of the top 20 most significantly enriched motifs in long-range limb enhancers.

The REX element contains two conserved LHX [C/T]AATTA motifs located within 30 bp from each other that are required for long range enhancer activity (Fig. 4a,c). To test whether having more than one [C/T]AATTA motif could be characteristic of other long-range enhancers,

we examined the number of [C/T]AATTA motifs in experimentally validated and multiome-predicted limb enhancers (Methods). Indeed, for limb enhancers containing more than one [C/T]AATTA motif the enhancer regions were on average 429 kb farther (median distance) from their target promoters than other limb enhancers ($P < 0.01$; Fig. 4e and Extended Data Fig. 6f–i). Taken together, these results indicate that conserved [C/T]AATTA HD motifs are enriched in long-distance limb enhancers throughout the genome and are frequently located within the conserved core of enhancers themselves (Extended Data Fig. 6k).

## LHX motifs are critical for ZRS activity

We next examined whether [C/T]AATTA motifs located inside enhancers are specifically required for their long-range enhancer activity. The ZRS enhancer region contains four previously uncharacterized [C/T]

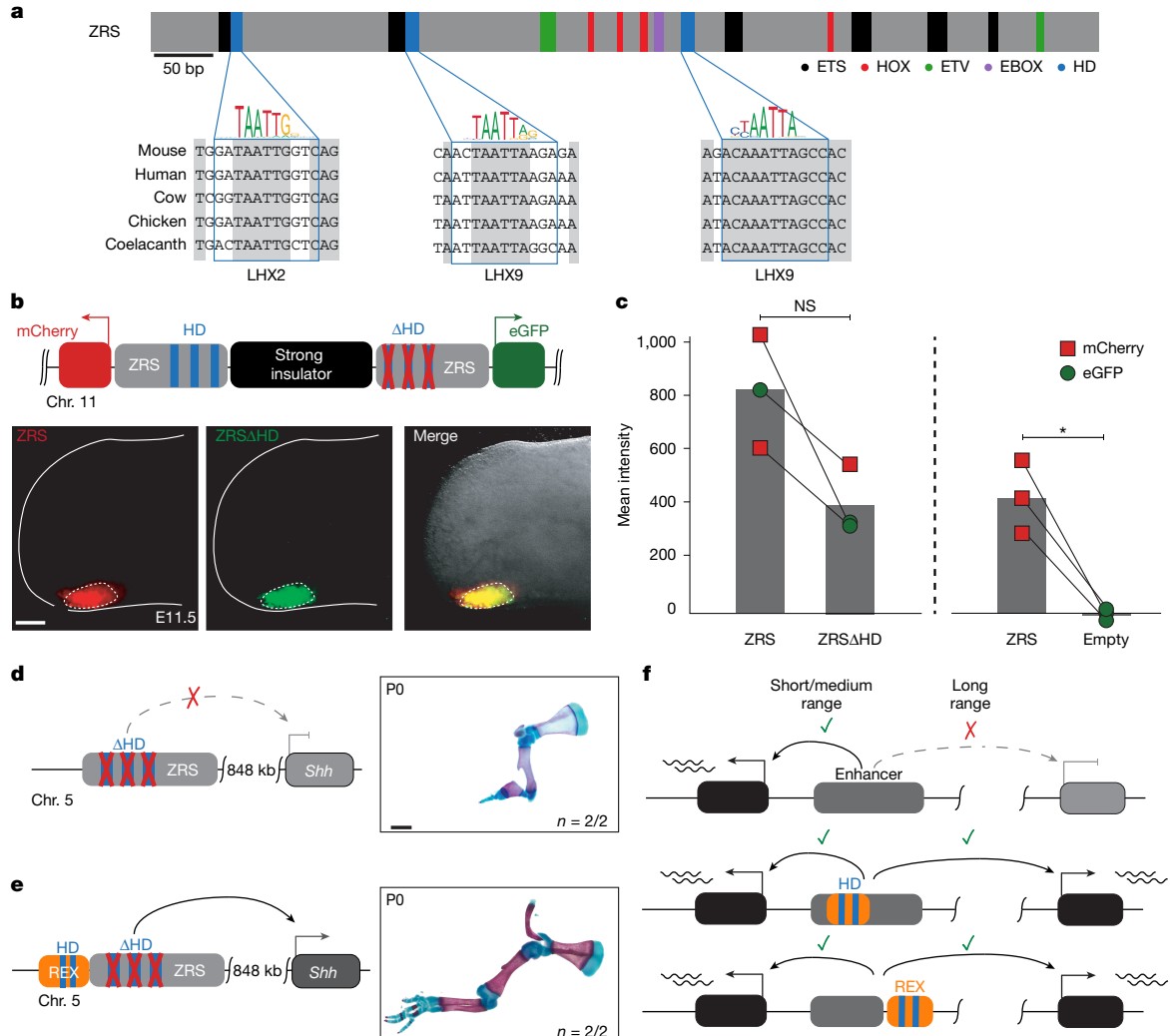

**Fig. 5 | [C/T]AATTA HD motifs are required for long-range gene activation.** **a**, The position of previously identified TF-binding sites within the mouse ZRS core (chromosome 5: 29314881–29315667; mm10)[45,70]. Conserved [C/T]AATTA motifs are highlighted in blue and their multispecies alignment is shown below. The blue boxes demarcate regions that were mutagenized. **b**, A dual-enSERT transgenic construct containing WT mouse ZRS driving *mCherry* (red) and the ZRS with mutated [C/T]AATTA HD motifs (ZRSΔHD) driving *eGFP* (green) separated by a strong synthetic insulator. Hindlimb bud images from a representative transgenic embryo are shown below. The white dotted line encircles the region that was quantified in **c**. **c**, The normalized mean fluorescence intensity in embryos containing the ZRS–mCherry/ZRSΔHD–eGFP or ZRSΔHD–mCherry/ZRS–eGFP (*n* = 3 embryos) and control ZRS–mCherry/Empty–eGFP (*n* = 3 embryos) constructs. Statistical analysis was performed using a two-sided paired *t*-test test with no adjustment for multiple comparisons; *\**P* = 0.0182. **d**, Skeletal forelimb preparation from E18.5 *ZRS^ΔHD/ΔHD* KI mice with three mutated [C/T]AATTA sites within the endogenous ZRS enhancer. **e**, Addition of the REX element to ZRSΔHD partially rescues limb outgrowth in *ZRS^ΔHD+REX/ΔHD+REX* KI. **f**, The proposed model of long-distance enhancer activation in the developing limb buds. Scale bars, 250 μm (**b**) and 1 mm (**d** and **e**).

AATTA motifs, three of which are located within the conserved core and are evolutionary conserved across all jawed vertebrates (Fig. 5a). To test their requirement for spatiotemporal limb-specific ZRS activity, we used dual-enSERT, our recently developed dual-fluorescence reporter system that enables comparison of enhancer allele activities in the same mouse[44]. We generated a construct in which a WT approximately 1.3 kb mouse ZRS drives *mCherry* and the same *ZRS* allele but with mutated [C/T]AATTA HD motifs (ZRSΔHD), drives *eGFP*. The transgenes were separated by a synthetic insulator to prevent reporter gene cross-activation. We left all other TF-binding sites that were previously shown to be important for ZRS enhancer activity intact, including ETS1, ETV, HOX and E-BOX motifs (Fig. 5a). We injected the resulting ZRS–mCherry/ZRSΔHD–eGFP bicistronic construct into mouse zygotes and collected transgenic embryos at E11.5. We detected mCherry and eGFP fluorescence in the ZPA, indicating that the ZRS enhancer lacking [C/T]AATTA motifs can act over short range and drive robust reporter expression (Fig. 5b,c and Extended Data Fig. 7c,d).

We obtained a similar result when we swapped fluorescent reporter genes, ruling out the influence of fluorophores on our observations (Fig. 5c). While both the ZRS and ZRSΔHD directed expression in a spatially highly restricted manner to the ZPA, ZRSΔHD showed lower, albeit not statistically significant, quantitative activity than the ZRS regardless of the fluorescent reporter pairing. This may be due to the close proximity of [C/T]AATTA HD motifs to the ETS sites, which are critical for ZRS activity[45]. Importantly, we observed only mCherry expression and no detectable eGFP expression in transgenic mice lacking enhancer upstream of the *eGFP*, indicating that there is no reporter cross-activation (*P* < 0.0182; Fig. 5c and Extended Data Fig. 7b).

Having established that disruption of [C/T]AATTA motifs does not abolish limb enhancer activity in a reporter assay, we examined whether these motifs are required for long-distance E–P communication. We created a KI mouse line in which we disrupted the same three [C/T]AATTA motifs within the endogenous ZRS enhancer region. Notably, disrupting the [C/T]AATTA motifs alone resulted in a loss of limb

outgrowth indistinguishable from complete loss of SHH function in the limb consistent with the specific requirement of [C/T]AATTA motifs for long-range ZRS activity (Figs. 1d and 5d). The loss of limb outgrowth in *ZRS^ΔHD/ΔHD* mice could be almost completely rescued by addition of the REX element; mice containing the ZRSΔHD allele fused to the REX element fully developed zeugopod and part of the autopod with four digits on both forelimbs and hindlimbs (Fig. 5e).

Taken together, our KI and transgenic results indicate that [C/T]AATTA HD motifs in the ZRS are dispensable for limb-specific activity at short range, but are critically required for long-range activity. These experiments uncouple the marked tissue specificity of the ZRS, directing highly restricted expression to the ZPA, from its ability to act over extremely long genomic distances. Our results also indicate that the loss of long-range enhancer activity after the removal of endogenous [C/T]AATTA HD motifs could be compensated by addition of a heterologous REX element, demonstrating that the [C/T]AATTA HD motifs found in the REX element and the ZRS are functionally equivalent.

## Discussion

Here we have identified an evolutionarily conserved signature that is specifically required to confer long-distance E−P communication at the *Shh* genomic locus. We also describe a prototypical element containing this signature, REX, that is sufficient to confer long-distance interactivity to heterologous short- and medium-range limb enhancers. This finding reveals a fundamental aspect of enhancer function that cannot be measured in traditional transgenic reporter experiments and provides a plausible mechanistic explanation for previous observations that linear distance can impact an enhancer's ability to activate gene expression[46–48].

*Cis*-regulatory elements that facilitate enhancer activity continue to emerge as essential regulators of gene expression. This list includes *Drosophila* tethering[49–51] and remote control elements[52] along with mammalian CTCF sites[28,53–57], CpG islands[58] and facilitator elements[59,60]. The REX element adds to this list but it does not share sequence similarity with any of the previously described elements. Both the REX element and the ZRS enhancer contain highly conserved [C/T]AATTA-binding sites that are required for long-range activation. Our experiments showing the preferential requirement of [C/T]AATTA motifs for long- but not short-range enhancer activity suggest that their cognate TFs could specifically mediate long-range E−P activation but may not be critical for general transcriptional activation. The [C/T]AATTA motifs match the binding preferences of many HD TFs, notably the LIM-HD TFs LHX2 and LHX9 that are required for normal limb outgrowth[61,62]. Indeed, LIM-domain-associated TFs such as GATA1, ISL1 and LHX2 are known to facilitate E−P looping at the β-globin locus as well as the assembly of interchromosomal olfactory gene compartments through their cofactor LIM-domain-binding factor (LDB1)[63–66]. However, their general role in regulating long-range E−P interactions is currently an area of open investigation[67,68]. Global enrichment of [C/T]AATTA HD motifs in long-distance enhancers suggests that there are probably more REX elements across the genome, with many of them integrated into the enhancer itself, as we showed with the ZRS (Fig. 5f).

As our enhancer-substitution experiments were performed at the *Shh* locus, we cannot rule out the potential impact of locus-specific effects on our findings. Further investigation into long- versus short-range enhancer activity at other genomic loci and across different cell types will be necessary to understand how widespread REX elements are in mammalian genomes and to dissect the molecular mechanism of REX element function. Nevertheless, our results suggest that cell-type specificity and long-range activity are two distinct aspects of gene activation by enhancers, probably fulfilled by separate sets of TFs. This separation will have implications for interpretation of human genetics studies as some non-coding variants will probably affect only long-range enhancer ability while others will disrupt cell-type-specific gene activation.

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

# Methods

## Ethics statement
The animal work conducted in this study was reviewed and approved by the Lawrence Berkeley National Laboratory Animal Welfare and Research Committee and University of California Irvine Laboratory Animal Resources (ULAR) under protocols AUP-20-001 and AUP-23-005. Mice were housed in the animal facility, where their conditions were electronically monitored continuously with daily visual checks by technicians.

## Mouse tissue collections
The FVB/NCrl strain *Mus musculus* (Charles River) was used for all breeding experiments and mouse embryonic tissue collection. Mice were mated using a standard timed breeding strategy and E10.5, E11.5, E13.5 and E18.5 embryos along with postnatal day 0 (P0) neonates of both sexes were collected for staining or dissection using approved institutional protocols. Embryonic litters were kept on ice and processed one at a time to avoid degradation during collection. Embryos at unexpected developmental stages were excluded from the study. Breedings were set up and litters collected until a minimum threshold was met (two embryos for phenotyping experiments and three embryos for statistical analysis related experiments) for each genotype of interest with all embryos acquired above that threshold processed as well. No randomization was needed for the experiments presented. The authors were blinded to embryo genotypes during embryo collection, and genotyping was performed afterwards.

## Generation of enhancer KI mice
All KI and transgenic targeting constructs in this study were generated using Gibson Assembly cloning (New England Biolabs). Sequences of all targeting constructs are provided in supplementary data. Before microinjections all targeting vectors were validated using whole-plasmid sequencing (Plasmidsaurus). KI mouse strains carrying replaced enhancer alleles were generated using a modified CRISPR–Cas9 pronuclear microinjection protocol[71,72]. KI enhancers were chosen on the basis of evidence of strong limb bud activity in reporter constructs[41,43,73,74].

The enhancers that were used for KI analysis were as follows: HS72 (chromosome 16: 51623899–51624805; hg38), MM1564 (chromosome 2: 60785660–60787563; mm10), HS1516 (chromosome 21: 38989433–38991200; hg38) and MM1492 (chromosome 4: 154707415–154711162; mm10). In brief, we used previously described sgRNA and donor vectors containing heterologous enhancers surrounded by homology arms, targeting the ZRS enhancer region (chromosome 5: 29314497–29315844; mm10) and new sgRNA and donor vector targeting the *Shh* promoter proximal region[71,72] (Extended Data Figs. 2a, 4d and 7e,f). KI mice were generated by injecting a mix of Cas9 protein (IDT, final concentration of 25 ng μl$^{-1}$), sgRNA (IDT, 50 ng μl$^{-1}$) and donor plasmids (7.5–25 ng μl$^{-1}$) in an injection buffer (10 mM Tris, pH 7.5; 0.1 mM EDTA) into the cytoplasm of FVB embryos according to standard procedures. Female mice (strain CD-1) were used as foster mothers. $F_0$ mice were genotyped using PCR and Sanger sequencing as previously described using primers outlined in Supplementary Table 4 (ref. 40). For the HS72 KI, the barcoded fragments were not included so primer pairs F1 + R8 and F6 + R7 were used instead of F1 + R1 and F2 + R2. Only founders carrying a single-copy KI at the ZRS locus were used for breeding[40]. A KI mouse strain carrying part of the *lacZ* sequence in place of the ZRS was previously generated[71].

## Generation of transgenic reporter mice
The HS72-Shh-promoter::lacZ construct was created by introducing the HS72 enhancer (chromosome 16: 51623899–51624805; hg38) into the hsp68-promoter-lacZ reporter vector[72] followed by replacement of the *hsp68* promoter with the mouse *Shh* promoter (chromosome 5: 28466764–28467284; mm10). The REX-Shh-promoter::lacZ construct was created by introducing the REX element into the PCR4-Shh-promoter::lacZ-H11 vector (Addgene, 139098)[40]. The sequence used for the REX element was extended to include the entire conserved block (Extended Data Fig. 5d) to more thoroughly test the enhancer activity of the fragment. The HS72-Shh-promoter::Shh construct was created by cloning the HS72 enhancer followed by the mouse *Shh* promoter and *Shh* ORF into the PCR4-Shh-promoter::lacZ-H11 vector in the opposite orientation relative to the *lacZ* reporter gene. ZRS−mCherry/Empty−eGFP, ZRS−mCherry/ZRSΔHD−eGFP and ZRSΔHD−mCherry/ZRS−eGFP were generated by inserting WT *ZRS* and *ZRSΔHD* sequences into the dual-enSERT-2.2 plasmid (Addgene, 211942)[44].

Transgenic mice carrying enhancer-reporter transgenes were created using random transgenesis (HS72-Shh-promoter::lacZ construct) or site-directed enSERT or dual enSERT transgenesis (REX-Shh-promoter::lacZ, HS72-Shh-promoter::Shh, and ZRS−mCherry/ZRSΔHD−eGFP and ZRSΔHD−mCherry/ZRS−eGFP constructs) as previously described[40,44,75,76]. After pronuclear microinjections (below), $F_0$ embryos were collected at E11.5 and processed for LacZ staining (HS72-Shh-promoter::lacZ and REX-Shh-promoter::lacZ), fluorescence imaging (ZRS−mCherry/ZRSΔHD−eGFP, ZRS−mCherry/Empty−eGFP and ZRSΔHD−mCherry/ZRS−eGFP) or at E13.5 to analyse limb morphology (HS72-Shh-promoter::Shh). Transgenic embryos were genotyped by PCR and Sanger sequencing as previously described[40,44]. The sequences of enhancer alleles and plasmids are provided in the Supplementary Information.

## Whole-mount in situ hybridization
The *Shh* transcript distribution in E10.5 and E11.5 mouse embryonic limb buds was assessed by whole-mount in situ hybridization as previously described[72]. E10.5 or E11.5 embryos were collected and fixed overnight in 4% paraformaldehyde (PFA), cleansed in PBT (PBS with 0.1% Tween-20) before being dehydrated through a methanol series to be preserved in 100% methanol. For in situ hybridization, the embryos were rehydrated, washed with PBT and then bleached with 6% $H_2O_2$/PBT for 15 min before being washed again with PBT. The samples were then permeabilized for 15 min with 10 μg ml$^{-1}$ of proteinase K in PBT, inactivated for 5 min in 2 mg ml$^{-1}$ glycine/PBT, rinsed twice with PBT and refixed for 20 min 0.2% glutaraldehyde/4% PFA in PBT. After fixation, the embryos were with PBT (×3, 15 min) and incubated in prehybridization buffer (50% deionized formamide, 5× SSC, pH 4.5, 2% Roche blocking reagent, 0.1% Tween-20, 0.5% CHAPS, 50 mg ml$^{-1}$ yeast RNA, 5 mM EDTA and 50 mg ml$^{-1}$ heparin) for 1 h at 70 °C, before overnight incubation in hybridization buffer containing 1 mg ml$^{-1}$ digoxigenin-labelled antisense riboprobes at 70 °C with gentle rotation. Dig-labelled riboprobes recognizing *Shh* were in vitro synthesized using the RNA Labelling Mix (Roche) and T3 RNA polymerase (Roche). Next, the embryos were washed multiple times in hybridization buffer with increasing concentrations of 2× SCC pH 4.5 finished with two washes with 2× SCC, 0.1% CHAPS. The embryos were next incubated for 45 min at 37 °C in 20 μg ml$^{-1}$ RNase A in 2× SSC, 0.1% CHAPS before being washed in maleic acid buffer (100 mM maleic acid disodium salt hydrate, 150 mM NaCl, pH 7.5) (twice for 10 min at room temperature and twice for 30 min at 70 °C). The embryos were then transitioned to TBST (140 mM NaCl, 2.7 mM KCl, 25 mM Tris-HCl, 1% Tween-20, pH 7.5) to prepare for an hour long blocking in 10% lamb serum/TBST and an overnight incubation at 4 °C in 1% lamb serum containing anti-Dig-AP antibody (Roche, 1:5,000). The next morning, the embryos were washed three times with TBST for 5 min to remove excess antibody and then further washed five times with TBST for 1 h before being equilibrated in NTMT (100 mM NaCl, 100 mM Tris-HCl, 50 mM $MgCl_2$, 1% Tween-20, pH 9.5). Alkaline phosphatase activity was detected by incubating in BM purple reagent (Roche) at room temperature in the dark with gentle agitation. The reaction was halted with five PBT washes for 10 min before transitioning to 4% PFA in PBS for long-term storage. The embryos were imaged on the Flexacam C1 camera mounted onto the Leica M125C stereomicroscope.

## Skeletal staining

Skeletal preparations were performed as previously described[72]. Embryos were collected at E18.5 or P0 and euthanized before overnight incubation in water at room temperature. The samples were then incubated in 65 °C water for 1 min before the epidermis and organs were removed. The samples were fixed in 95% ethanol for storage before staining according to a standard Alcian Blue/Alizarin Red protocol[77]. The samples were incubated for at least 24 h in Alcian Blue stain (15% Alcian Blue, 20% acetic acid in ethanol) followed by three washes with ethanol and an overnight ethanol incubation. Next, the samples were cleared in 1% KOH for 20 min followed by counter-staining with Alizarin Red (5% Alizarin Red in 1% KOH) for 4 h. The embryos were then further cleared with 1% KOH for 15 min followed by incubation in decreasing concentrations of 1% KOH and glycerol. The stained embryos were dissected in 80% glycerol and limbs were imaged at 1× using the ZEISS Stemi 508 microscope and the Axiocam 208 digital camera.

## LacZ Staining

LacZ staining was conducted as previously described[40]. The embryos were collected at E11.5 and fixed in 4% PFA for 30 min before washing three times for 30 min in embryo wash buffer (2 mM MgCl₂, 0.01% deoxycholate, 0.02% NP-40, 100 mM phosphate buffer, pH 7.3). The embryos were then stained overnight in X-gal staining solution (0.8 mg ml⁻¹ X-gal, 4 mM potassium ferrocyanide, 4 mM potassium ferricyanide, 20 mM Tris, pH 7.5 in wash buffer) to visualize LacZ activity. The next morning, the embryos were rinsed with PBS three times for 10 min and fixed again in 4% PFA. Images were recorded on the ZEISS Stemi 508 microscope with the Axiocam 208 digital camera.

## qPCR

The anterior and posterior parts of hindlimbs from $Shh^{HS72(+2kb)/WT}$ E11.5 embryos were dissected from each individual embryo. The tissue was dissociated with collagenase for 10 min at 37 °C, followed by three washes with 500 µl 0.04% BSA in PBS, resuspended in RNAProtect (Qiagen, 76104) and frozen at −80 °C. To isolate RNA, the samples were pelleted at 4 °C, lysed and column-extracted using the RNeasy Kit (Qiagen, 74104). The samples were DNA digested in-column using DNase I (NEB, M0303S). cDNA was generated using the ProtoScript First Strand cDNA Synthesis Kit (NEB, E6300S). qPCR reactions were performed using the iTaq Universal SYBR Green Supermix (Bio-Rad, 1725121) using the C1000 Touch Thermal Cycler (Bio-Rad, 1845096). Then, .zpcr files were processed using the CFX Maestro Software (Bio-Rad) to extract cycle threshold values. *Shh* expression levels were normalized to the *Gapdh* housekeeping gene. Data were analysed using the rstatix package in R, with a Wilcoxon test used to determine statistical significance.

## ATAC–seq library construction, sequencing and data analysis

Posterior limb bud tissue containing the ZPA region from E11.5 mouse embryos was dissected and manually dissociated, followed by snap-freezing. After embryo genotyping, heterozygous embryos were processed for ATAC–seq. Library construction was performed using the Omni-ATAC–seq protocol[78,79], with an adjusted lysis step and transposition buffer based on the Omni-ATAC protocol[80]. For the transposition reaction, nuclei were treated with Illumina Nextera transposase. The reaction was cleaned up using the Zymo DNA Clean and Concentrator kit, followed by preamplification of the transposed DNA using Nextera indexed primers with 5 cycles of PCR. The number of additional PCR cycles was determined by qPCR using 5 µl of a partially amplified library. The resulting library was cleaned using AMPure XP beads (Beckman Coulter) and quantified by qPCR using the Kapa Sybr Fast universal for Illumina Genome Analyzer kit. The library size was determined using the Bioanalyzer 2100 DNA High Sensitivity Chip (Agilent). The library was sequenced on the Illumina NovaSeq 6000 system using 100 cycles and paired-end dual-index read chemistry. The version of

NovaSeq control software used was NVCS v.1.6.0 with real-time analysis software RTA v.3.4.4.

Fastq files were trimmed and clipped using trim_galore (https://www.bioinformatics.babraham.ac.uk/projects/trim_galore/) using the following options: -q 20 --stringency 2 --clip_R1 16 --clip_R2 16. Reads were then aligned to the original mm10 genome assembly and a modified version of the genome containing the corresponding insertion using bedtools getfasta[81]. As the human HS72 enhancer inserted on chromosome 5 has an orthologous region on chromosome 8 of the mouse genome, we also mapped the reads only to chromosome 5 of the mm10 genome containing the insertion and only to chromosome 8 of the mm10 genome and could verify that no reads coming from the insertion were lost at the mouse orthologous region. Reads mapping to the positive strand were shifted by +4 bp and reads mapping to the negative strand were shifted by −5 bp to correct for the Tn5 insertion bias using the Unix command awk. Genomic tracks were generated using bedtools genomecov[81] by normalizing the mapped read counts to 1 million and dividing the read counts by two, except at the insertion sites.

## Single-cell multiomics and data analysis

Single-cell multiome ATAC–seq/RNA-seq was performed using a modified 10x protocol (10x Genomics, protocol CG000169 Rev D). WT mouse embryos were collected at E11.5. A single hindlimb bud was dissected in ice-cold PBS and incubated with collagenase II (Gibco, 17101015, 0.2 µl at 100 U µl⁻¹) for 10 min at 37 °C. Every 5 min, hindlimb tissue was triturated using a P200 pipette for mechanical dissociation into single cells. Immediately after collagenase treatment, 10% FBS (450 µl) was added and dissociated cells were centrifuged. The supernatant was removed, and the cells were resuspended in 100 µl of ice-cold 10× nuclear lysis buffer (10x Genomics, protocol CG000169 Rev D) and incubated for 5 min on ice. Ice-cold wash buffer was added to the cells, lysed cells were centrifuged and the supernatant was removed. Nuclei were resuspended in an ice-cold wash buffer, quantified and inspected for viability using the Trypan Blue assay (Bio-Rad, 1450013). Nuclei were loaded at a concentration that would enable recovery of 10,000 nuclei using the 10x Chromium Single Cell Multiome ATAC + Gene Expression kit (10X Genomics, 1000285). Paired-end sequencing was performed on the Illumina NovaSeq 6000 system for approximately 52,000 and 26,000 reads per cell for RNA-seq and ATAC–seq, respectively.

Fastq files were aligned to the mm10 genome assembly and barcodes were counted using CellRanger ARC (v.2.0.2)[82]. Expression counts and chromatin peak matrices generated from CellRanger were further processed using the Signac R package[83]. The snATAC–seq dataset included a median of 31,000 reads and 14,200 high-quality fragments per cell, while the snRNA-seq dataset comprised a median of 68,000 unique molecular identifiers (UMIs) and 4,300 genes detected per cell. Low-quality nuclei were filtered out using standard Signac parameters and MACS2 was used for peak calling[84]. Transcriptome data were normalized and dimensionality was reduced using PCA (Seurat). DNA accessibility data were normalized using latent semantic indexing (Seurat). From the combined Seurat object, UMAP and nearest-neighbour analyses were performed to identify clusters with a resolution of 1 across 20 dimensions, as indicated by the ElbowPlot function, and based on their chromatin and gene expression profiles. Cell identities for the resulting 16 clusters were assigned using well-known marker genes from the literature and past scRNA-seq datasets[85] (Extended Data Fig. 1 and Supplementary Table 2).

## E–P assignment

To link limb enhancers to their putative target genes, gene expression and open chromatin peaks (distance of 10 kb to 2 Mb) were correlated using the LinkPeaks function from Signac[83]. This function accounts for bias in GC content (as Tn5 transposase is inherently biased towards cutting GC-rich regions), overall accessibility and peak size. For analysis

in Figs. 1 and 4 and Extended Data Fig. 6, we used enhancers that were open (accessibility > 1) and genes that were expressed (normalized expression > 0.25) in mesenchymal cells.

To assign putative target genes to MM1492, mHS1516m mHS72 and MM1564 enhancers that were used in KI experiments, we additionally used enhancer capture Hi-C data[71]. We further validated these putative target genes by comparing enhancer activity patterns and gene expression patterns from previously published whole-mount in situ hybridization experiments for *Prmd16*[86], *ETS2*[86], *Rbms1*[87] and *Sall1*[88] to ensure that expression of the target gene overlapped with the region of enhancer activity.

## Motif enrichment analysis

A list of functionally validated limb enhancers was obtained from the VISTA enhancer browser[43]. To assign putative target genes for these enhancers, we used the multiome E–P assignments or enhancer capture Hi-C data[71]. Putative target genes were restricted to limb-expressed genes (curated from the hindlimb scRNA-seq (this study) and whole-limb bulk RNA-seq (Limb-Enhancer Genie[89]). For enhancers with multiple target genes, we considered the longest (or shortest; Extended Data Fig. 6) distance interaction followed by separation into two groups based on the distance between the enhancer and target gene: short-range (10 kb to 200 kb) and long-range (400 kb to 2 Mb). We then performed differential motif enrichment analysis using the findMotifsGenome.pl command in HOMER with a given size[90] comparing short- and long-range E–P sets for (1) predicted E–P pairs for bona fide enhancers (using VISTA enhancer coordinates) and (2) predicted E–P pairs for putative limb enhancers defined by scATAC–seq. HOMER and JASPAR2022[91] motifs were used in the motif search. Only motifs for limb-expressed TFs, from the same list used to filter target genes, were considered in the analysis.

To detect [C/T]AATTA HD motifs within enhancers (Figs. 4 and 5 and Extended Data Fig. 6), we used MA0700.2.LHX2 and LHX9 (Homeobox)/Hct116-LHX9.V5-ChIP-Seq (Gene Expression Omnibus: GSE116822) position weight matrices (PWMs) and the findMotifsGenome.pl command from HOMER. This was followed by filtering conserved motifs using the Genomic Scores package[92] and phastCons60way.UCSC.mm10 dataset to an average conservation score of greater than zero. For the motif occurrences in Extended Data Fig. 6k, the conservation of each motif occurrence was analysed using the Table Browser[93] tool with 100 Vert. Cons (phyloP100wayAll) for human sequences or Vertebrate Cons (phyloP30wayAll) for mouse sequences. Overlapping motif occurrences were collapsed into one region for subsequent analysis. Statistical significance was calculated using a Wilcoxon test.

## Motif mutagenesis in transgenic and KI mice

The [C/T]AATTA HD motif regions within the ZRS were replaced with random sequences. FIMO[94] was used to ensure that the mutagenized sequence does not have any [C/T]AATTA motif matches. Changes were as follows: GGATAATTGGTC>CGACGTCTGTAG (first HD site), AACTAATTAAGA>TCGACGGACACT (second HD site) and GACAAATTAGCC>AGGGACTGCTCT (third HD site). The resulting approximately 1.3 kb ZRSΔHD sequence was used for dual-enSERT and KI analysis (Fig. 5 and Extended Data Fig. 7). The presence of motifs in KI mice was confirmed through PCR (Supplementary Table 4 and Extended Data Figs. 2a and 7e,f) and Sanger sequencing.

For the REX element mutagenesis, the [C/T]AATTA HD motif regions or the LEF1 motif were replaced with random sequences within the KI targeting vector (Supplementary Information). Changes were as follows: ATCCTAATTGAAG>CCGACGTCTGTAG (first HD site) and ATTTGATTAGC>TTACAGCCCAG (second HD site) or ACTTTCATC>CTGTCGTCG (LEF1 site). Sequence changes in KI mice were confirmed using sanger sequencing and PCR using the primers F1, R1, F2 and R2.

## Live fluorescence imaging and quantification

Imaging of dual-enSERT embryos was conducted as previously described[44]. Embryos were imaged using the Zeiss V20 stereoscope equipped with a monochromatic camera (Axiocam 202, Zeiss), fibre optic light source (Zeiss, CL1500) and LED fluorescent laser (X-Cite, Xylis) emitting at 488 nm (eGFP) and 555 nm (mCherry) wavelengths. Merged images were created using the Zeiss Biolite Software.

For quantification, .czi files were imported into Fiji[95]. The region of interest, the ZPA, was defined around the brightest region of mCherry signal and applied to the GFP channel. A control region in anterior limb bud was also delineated to determine the background fluorescence of the embryo in each channel. The mean intensity was measured within both regions for both channels. The measurement for the background was subtracted from the region of interest to generate the mean intensity. For statistical analysis comparing the strength of the ZRS versus ZRSΔHD, the embryos with ZRS–mCherry/ZRSΔHD–eGFP ($n = 2$) and ZRSΔHD–mCherry/ZRS–eGFP ($n = 1$) were combined into one dataset ($n = 3$).

## Statistics and reproducibility

No previous analyses were used to determine sample size before experimentation. Embryos that were not at the correct developmental stage were excluded from data collection. For LacZ and skeletal Alcian Blue/Alizarin Red staining, embryos were identified by number and the researchers were blinded to the genotype. Statistical analysis was performed using the rstatix package. To determine the statistical significance of [C/T]AATTA motif enrichment and to compare the significance of the expression of *Shh* by qPCR, Wilcoxon rank-sum tests were used. For comparison of fluorescence intensity, significance was assessed using a two-sided pairwise *t*-test.

## Reporting summary

Further information on research design is available in the Nature Portfolio Reporting Summary linked to this article.

## Data availability

The scATAC–seq/scRNA-seq (Multiome) sequencing data generated in this study are available at GEO (GSE243635). The previously published capture Hi-C data[71] that were used in the analysis are available at GEO (GSE217078). The H3K27ac data were downloaded from the GEO (GSE108880). The CTCF-binding data were downloaded from the GEO (GSE84795). Transgenic embryo images from short-range enhancer reporter assay experiments were acquired from the VISTA enhancer browser[43].

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

**Acknowledgements** We thank the members of the Chao Family Comprehensive Cancer Center Transgenic Mouse Facility, Genomics High Throughput Facility and Optical Biology Core Facility Shared Resources for support; A. Syed and S. Sun for assistance with microscopy and image analysis software; and S. Lall and Kvon laboratory members for comments and suggestions on the manuscript. This work was supported by National Institutes of Health grants DP2GM149555 and R01HD115268 (to E.Z.K.), R01HG003988 (to L.A.P.), T32NS082174, T32GM008620 (supporting E.W.H.), GAANN Fellowship P200A220015 (supporting J.A.A.) and Ruth L. Kirschstein Predoctoral Individual NRSA fellowships F31HD112201 (to G.B.) and F30HD110233 (to E.W.H.). The Chao Family Comprehensive Cancer Center is supported in part by the National Cancer Institute of the NIH under award number P30CA062203. J.L.-R. is supported by the Spanish MICINN through the PID2023-148267NB-I00 grant and the CEX2020-001088-M institutional grant. D.S.-K. is supported in part by an unrestricted grant from Research to Prevent Blindness to the Gavin Herbert Eye Institute at the University of California. Research conducted at the E.O. Lawrence Berkeley National Laboratory was performed under Department of Energy Contract DE-AC02-05CH11231, University of California. This work was made possible, in part, through access to the Genomics High Throughput Facility Shared Resource of the Cancer Center Support Grant (CA-62203) at the University of California, Irvine and NIH shared instrumentation grants 1S10RR025496-01, 1S10OD010794-01 and 1S10OD021718-01. The content is solely the responsibility of the authors and does not necessarily represent the official views of the National Institutes of Health.

**Author contributions** E.Z.K. conceived the project with input from G.B., D.E.D., A.V. and L.A.P.; G.B. and E.Z.K. designed experiments. G.B. and E.Z.K. performed enhancer KI and transgenesis studies with help from S.H.J., B.C., K.C., M.L. and A.D.; G.B. performed ATAC–seq experiments. Q.X. and D.S.-K. assisted with developing ATAC–seq experiments. A.F.B. analysed the ATAC–seq data. E.W.H. performed 10x multiome experiments and analysed the data. E.W.H. and G.B. performed motif analysis. J.A.A. performed and analysed qPCR data. A.A.-C. and J.L.-R. performed in situ hybridization experiments. E.Z.K and G.B. wrote the manuscript with input from all of the other authors.

**Competing interests** The authors declare no competing interests.

**Additional information**
**Correspondence and requests for materials** should be addressed to Evgeny Z. Kvon.

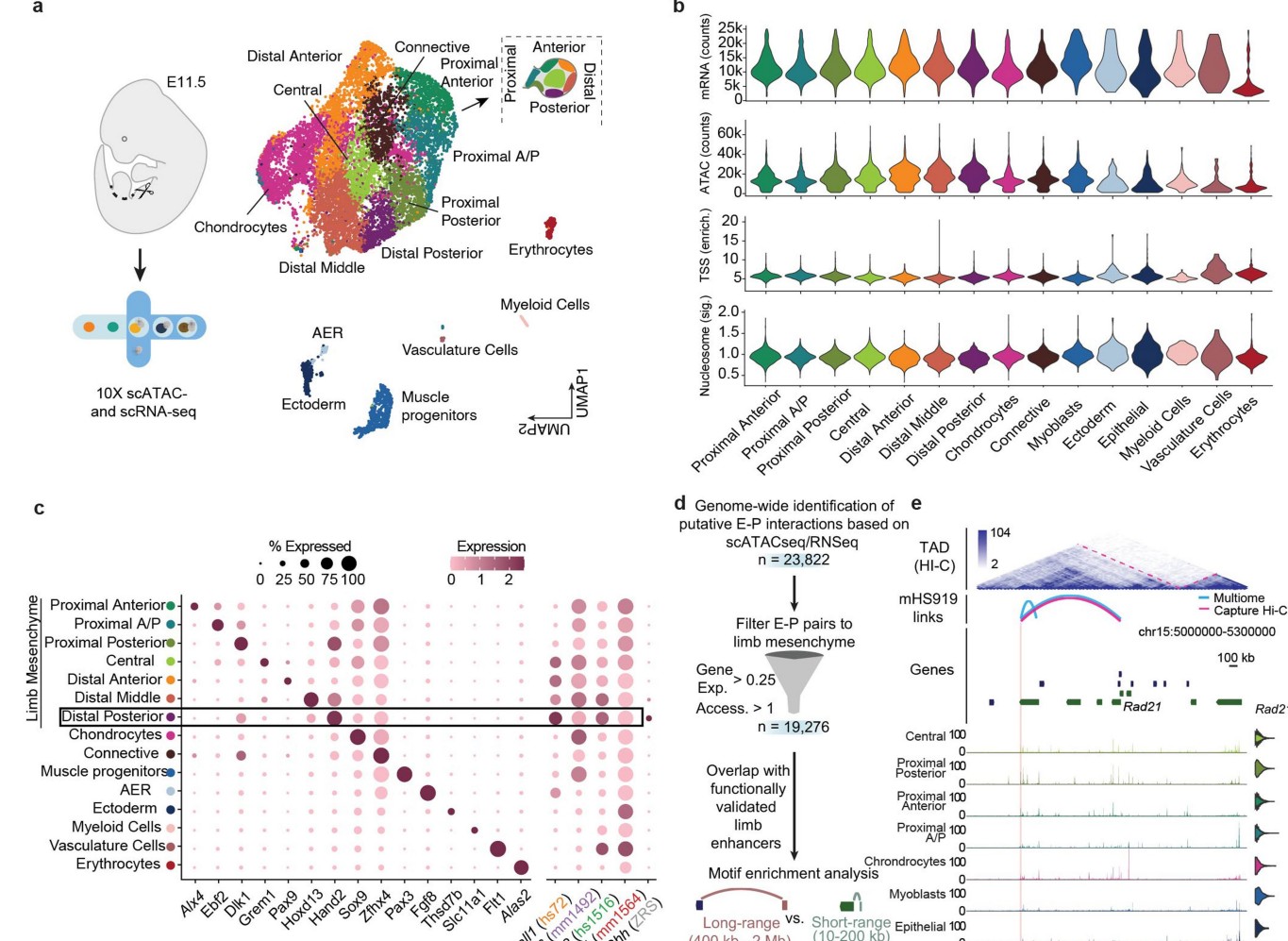

**Extended Data Fig. 1 | Multiome analysis (scATAC-seq + scRNAseq) in a mouse hindlimb bud.** (**a**) Single-cell multiomics in developing E11.5 mouse hindlimb bud. The UMAP plot depicts cell type clusters based on integrated scRNA-seq and scATAC-seq data, along with a cartoon mapping the different regions of the E11.5 hindlimb bud. (**b**) Single-cell multi-omics quality control violin plots for a number of mRNA counts, ATAC read counts, transcription start site (TSS) enrichment, and nucleosome signal by each cell type cluster. (**c**) Dot plot showing expression of cell-type-cluster-specific genes (left) and putative target genes of enhancers tested in this study (right) across cell clusters. Colour represents expression level and the size of the circle depicts

the percent of cells expressing each gene. The boxed region (Distal Posterior) encompasses the posterior limb bud region containing the Zone of Polarizing Activity (ZPA), where the ZRS normally activates *Shh*. Apical ectodermal ridge (AER). (**d**) Schematic pipeline for genome-wide identification of putative enhancer-promoter interactions in the hindlimb. (**e**) Example of predicted E-P interaction between mHS919 limb enhancer and *Rad21*. The *Rad21* locus (chr15:5000000-5300000; mm10) is shown with Hi-C data (top)[96], pseudobulk chromatin accessibility tracks and a violin plot for *Rad21* expression by cell type (bottom). Arches indicate mHS919 enhancer-centric E–P interactions from enhancer capture Hi-C (red) or multiome (blue).

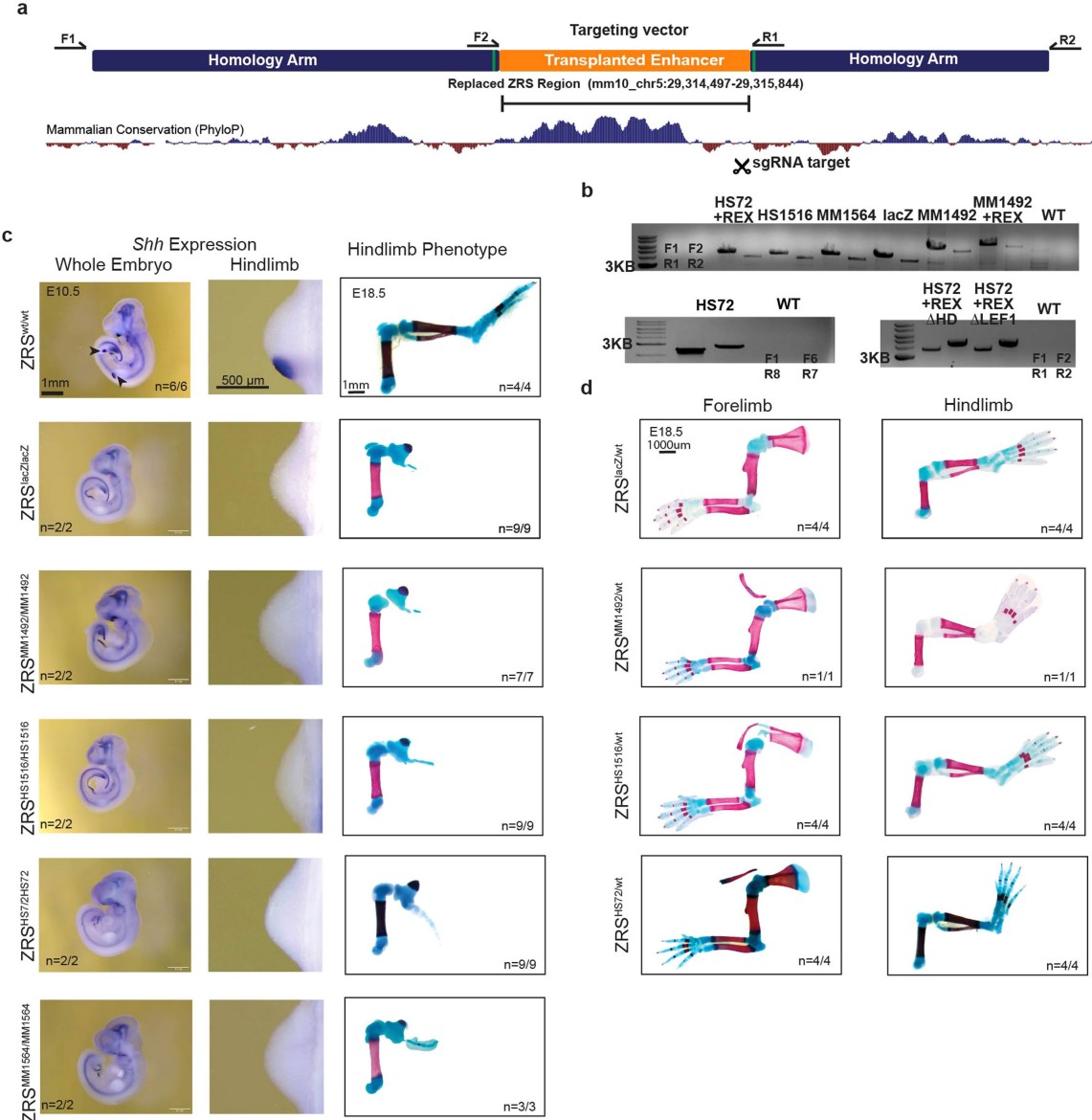

**Extended Data Fig. 2 | Generation and characterization of mice with transplanted enhancers.** (**a**) Schematic overview of enhancer replacement strategy. A 4.5 kb mouse genomic region containing the ZRS enhancer is shown together with the vertebrate phyloP conservation (dark blue). The donor vector contained two homology arms with vector-specific sequences for genotyping (green) and a corresponding replaced region containing the transplanted enhancer and mutagenized sgRNA recognition site (black, 5'-agtaccatgcgtgtgtTtTagCC-3'). PCR primers used for genotyping are shown as arrows. See Methods and Kvon et al.[40] for more details. (**b**) Shown are the results of PCR genotyping for each knock-in mouse line. One representative sample is shown for each genotype consistent with the results seen for all embryos and mice of the lines represented that were analysed in this study. For gel source data, see Supplementary Fig. 1. (**c**) *Shh* mRNA whole mount in situ hybridization analysis in wild type and knock-in mouse embryos (first two columns). The corresponding hind limb skeletal preparations of E18.5 wild type and knock-in mouse embryos are shown (third column). The number of embryos that exhibited representative limb phenotype over the total number of embryos with the genotype is indicated. Genotypes for each row are displayed on the left. (**d**) E18.5 skeletal staining showing the forelimb (left panel) and hindlimb (right panel) phenotype in heterozygous knock-in embryos.

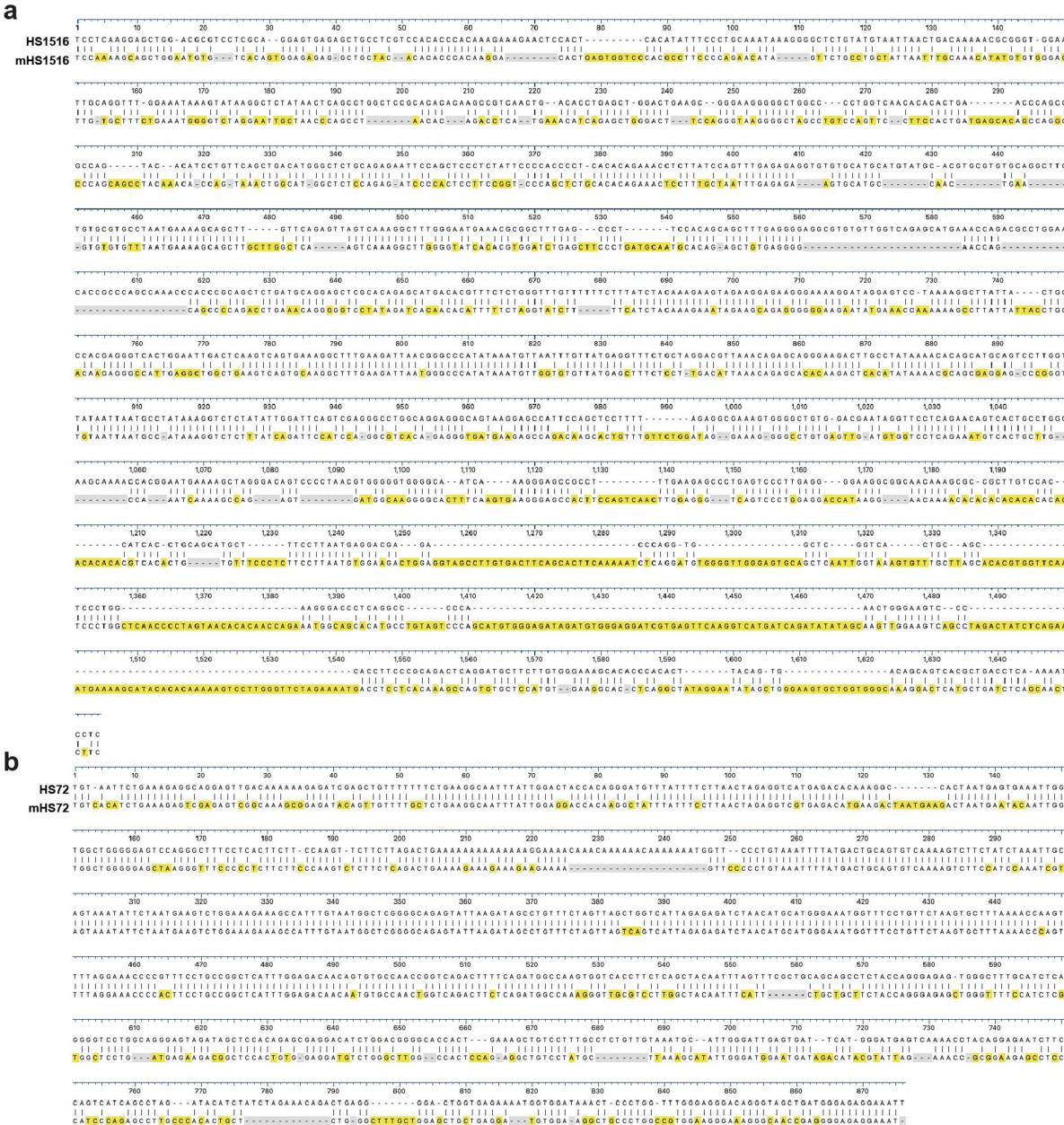

**Extended Data Fig. 3 | Pairwise alignment of the human HS72 and HS1516 enhancers.** Pairwise sequence alignments of the HS1516 (**a**) and HS72 (**b**) enhancers with their respective mouse homologues are shown.

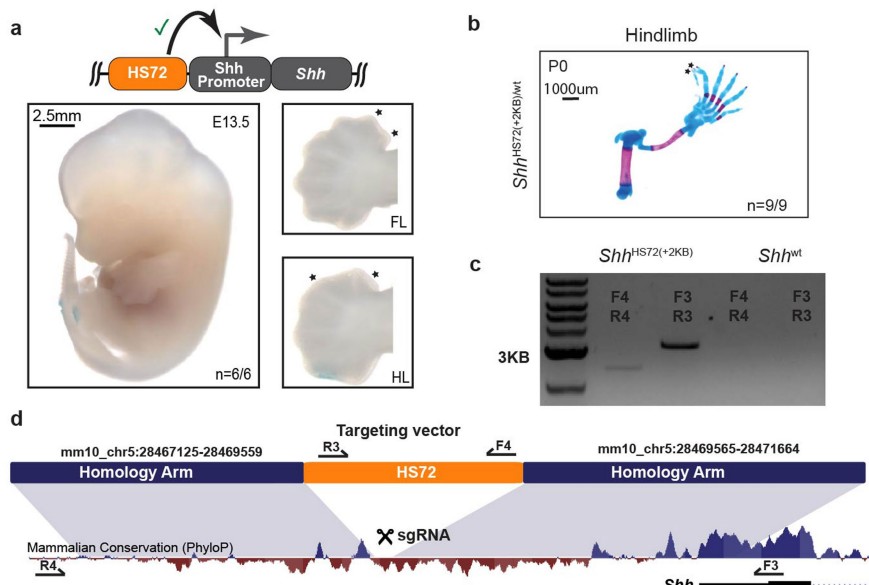

**Extended Data Fig. 4 | The HS72 enhancer can activate *Shh* at short-range.**
(**a**) Transgenic E13.5 mouse embryo with the HS72 limb enhancer placed upstream of the *Shh* promoter and ORF and integrated at H11 safe-harbour locus (light blue). Close-up images of limbs are shown on the right. Asterisks indicate extra digits (polydactyly). Numbers of embryos with limb polydactyly in both forelimb and hindlimb buds over the total number of transgenic embryos screened are shown. FL, forelimb. HL, hindlimb. (**b**) Hindlimb skeletal staining of P0 *Shh*^HS72(+2KB/wt)^ mice. (**c**) PCR gel electrophoresis confirming the correct integration of the left and right homology arms in the *Shh*^H272(+2KB)^ line.

One representative sample is shown for each genotype consistent with the results seen for all embryos and mice analysed in this study. (**d**) Model of the targeting construct used to insert the HS72 enhancer approximately 2 KB upstream of *Shh*. The donor vector contained two homology arms and a corresponding replaced region containing the transplanted enhancer and removing the sgRNA recognition sites (purple, 1: 5'-gggatcatgaggctggccacAGG-3' and 2: 5'-aaaggccacatttcttcctgTGG-3'). PCR primers used for genotyping are shown as arrows. For gel source data, see Supplementary Fig. 1. See Methods for more details.

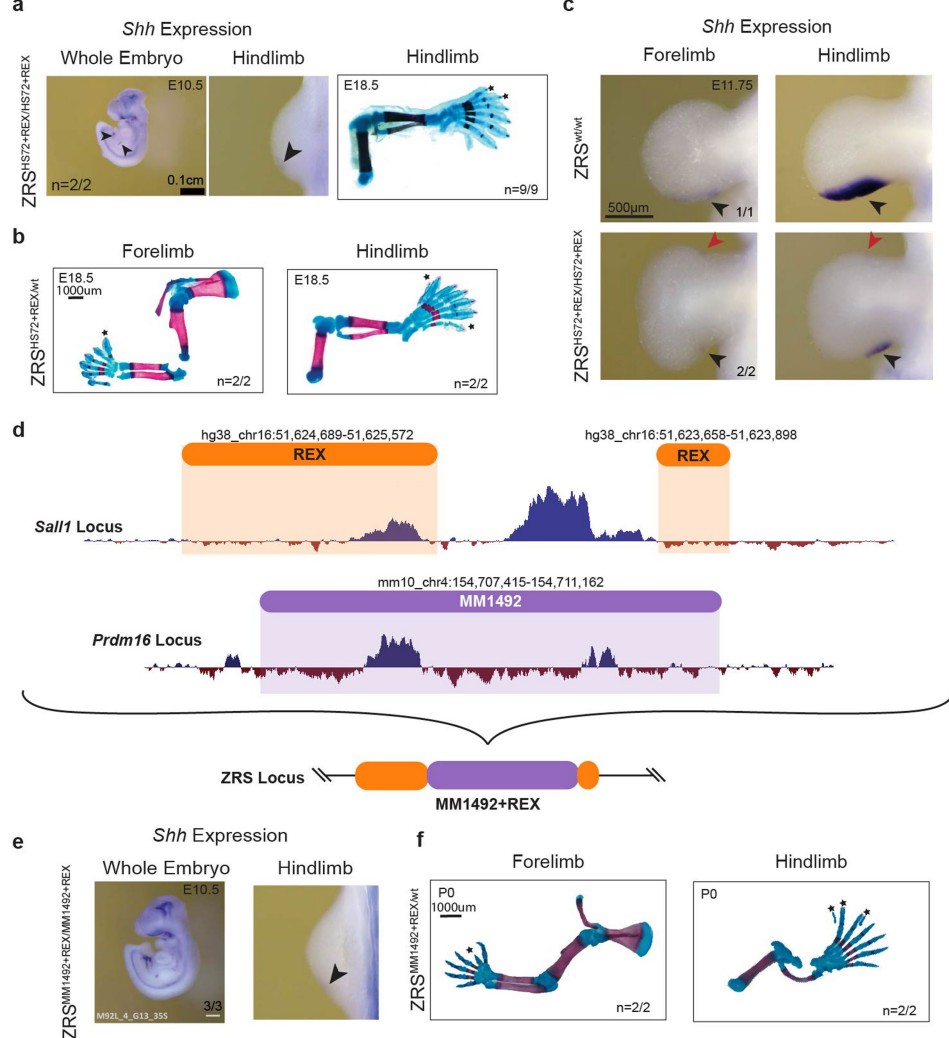

**Extended Data Fig. 5 | The REX element is required and sufficient for long-range activity at the *Shh* locus.** (**a**) *Shh* gene expression analysis using mRNA whole mount in situ hybridization in homozygous *ZRS^HS72+REX/HS72+REX* knock-in mouse embryos at E10.5 (left two panels). Arrows point to the *Shh* expression in the ZPA. The corresponding hindlimb skeletal preparations at E18.5 are shown (third column). The number of embryos that exhibited representative limb phenotype over the total number of embryos with the genotype is indicated. (**b**) Forelimb (first panel) and hindlimb (second panel) skeletal phenotypes in heterozygous *ZRS^HS72+REX/wt* knock-in embryos at E18.5. (**c**) Comparative *Shh* mRNA in situ hybridization analysis in wild type (top row) and homozygous *ZRS^HS72+REX/HS72+REX* knock-in (bottom row) mouse embryos during limb bud development in E11.75 embryos. Black arrows point to areas of *Shh* expression

in the ZPA. Red arrows point to ectopic expression in the anterior portion of the limb bud. * Extra digits. (**d**) Schematic showing the sequences from the HS72 (chr16:51623658-51623900 and chr16:51624689-51625572; hg38) and MM1492 (chr4:154706480-154712089; mm10) loci that make up the chimeric MM1492 + REX element. The orange bars indicate the expanded region flanking the core HS72 enhancer that contains the REX element. The purple bar marks the MM1492 enhancer sequence. The core HS72 enhancer was swapped out for the MM1492 enhancer to generate the MM1492 + REX construct (bottom). (**e**) *Shh* gene expression analysis using mRNA whole mount in situ hybridization in homozygous *ZRS^MM1492+REX/MM1492+REX* knock-in mouse embryos at E10.5. (**f**) Forelimb and hindlimb skeletal phenotypes of heterozygous *ZRS^MM1492+REX/wt* embryos at P0.

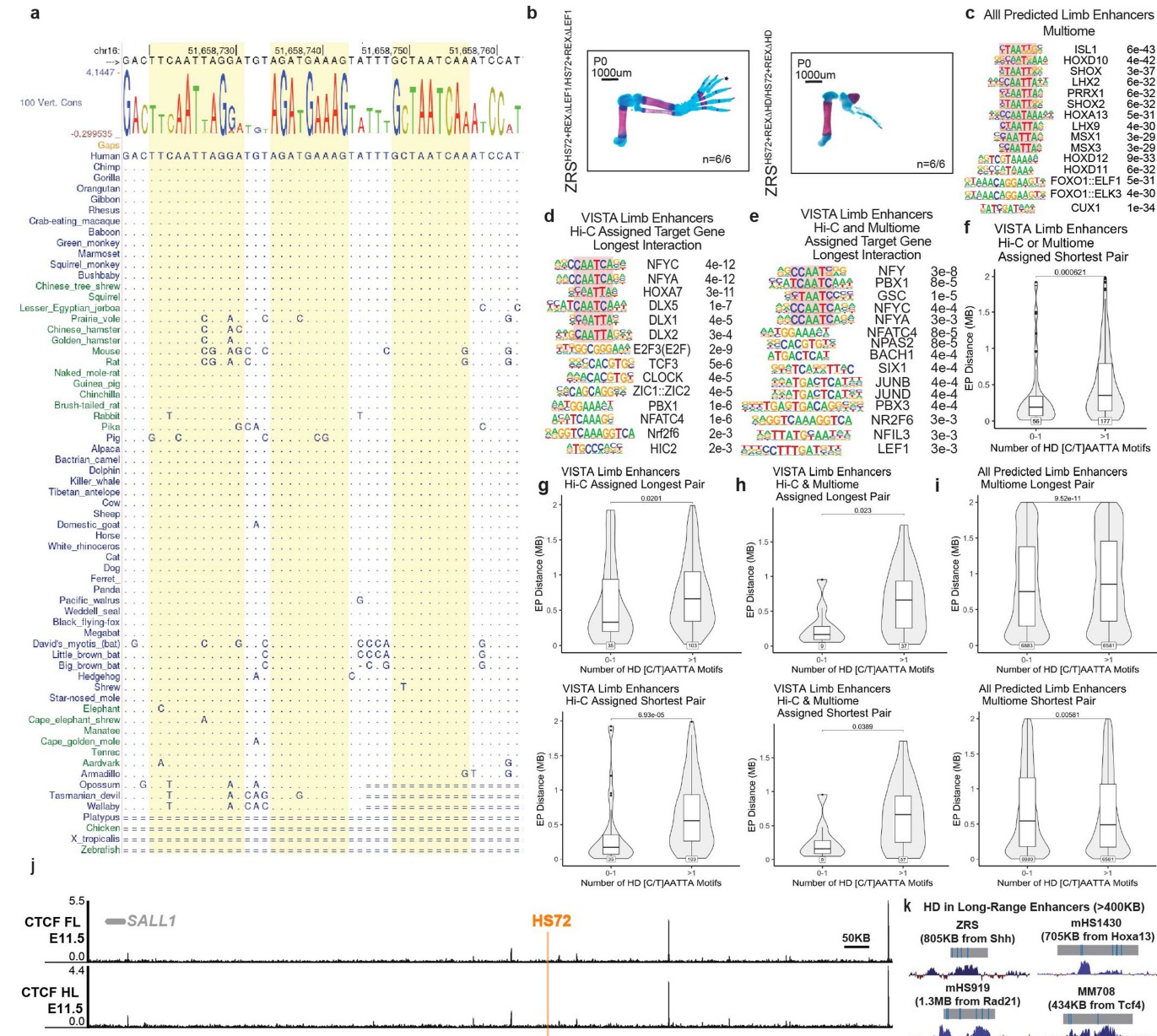

**Extended Data Fig. 6 | Putative TF binding sites in the REX element and other limb enhancers.** (**a**) Multiple sequence alignment (from UCSC genome browser) across 64 vertebrate species showing conserved LHX2, LEF1, and LHX9 TF motifs in the REX element. (**b**) Hind limb skeletal staining from *ZRS*[HS72+REXΔLEF1/HS72+REXΔLEF1] and *ZRS*[HS72+REXΔLHX/HS72+REXΔLHX] P0 mice. (**c-e**) Motifs enriched in long-range (400 kb - 2MB) compared to short-range (10 kb - 200 kb) enhancers assigned to their longest interacting target gene. In (**c**) all enhancer regions predicted by scATAC-seq/scRNA-seq were used while in (**d**) target genes were assigned to experimentally validated limb enhancers[43] by Hi-C while in (**e**) target genes were only assigned to experimentally validated limb enhancers[43] if they were identified by both Hi-C and scATAC-seq/scRNA-seq (Multiome). (**f-i**) Distribution of enhancer-promoter distances for putative enhancers with 0-1 compared to more than 1 [C/T]AATTA HD motifs. The box plots show the interquartile range (IQR; Q1–Q3) (box edges), the median (middle line), and the minimum (Q1 − 1.5 × IQR) and maximum (Q3 + 1.5 × IQR) values (whiskers). P-values are reported on the charts from a two-sided

Wilcoxon rank sum test with no adjustments for multiple comparisons. (**f**) For experimentally validated limb enhancers assigned to their nearest interacting target gene by Hi-C or Multiome (n = 233 independent enhancers-promoter pairs). (**g-i**) Assignment to the longest target identified is shown in the top panel, while shortest is shown in the bottom. (**g**) Shows experimentally validated limb enhancers assigned a target gene through Hi-C (n = 138 independent enhancers-promoter pairs). (**h**) Shows experimentally validated limb enhancers where the target gene was assigned by both Hi-C and scATAC-seq/scRNA-seq (Multiome) (n = 46 independent enhancers-promoter pairs) (**i**) Shows all predicated limb enhancers assigned target genes by Multiome (n = 13,464 independent enhancers-promoter pairs). (**j**) CTCF ChIP-seq from E11.5 fore- and hindlimb buds[97] showing CTCF binding at *Sall1* locus. HS72 and the REX element are highlighted in orange. (**k**) Shows the location of [C/T] AATTA HD motifs called by HOMER in representative long-range limb enhancers assigned the same target gene by both Hi-C and scATAC-seq/scRNA-seq together with evolutionary conservation track.

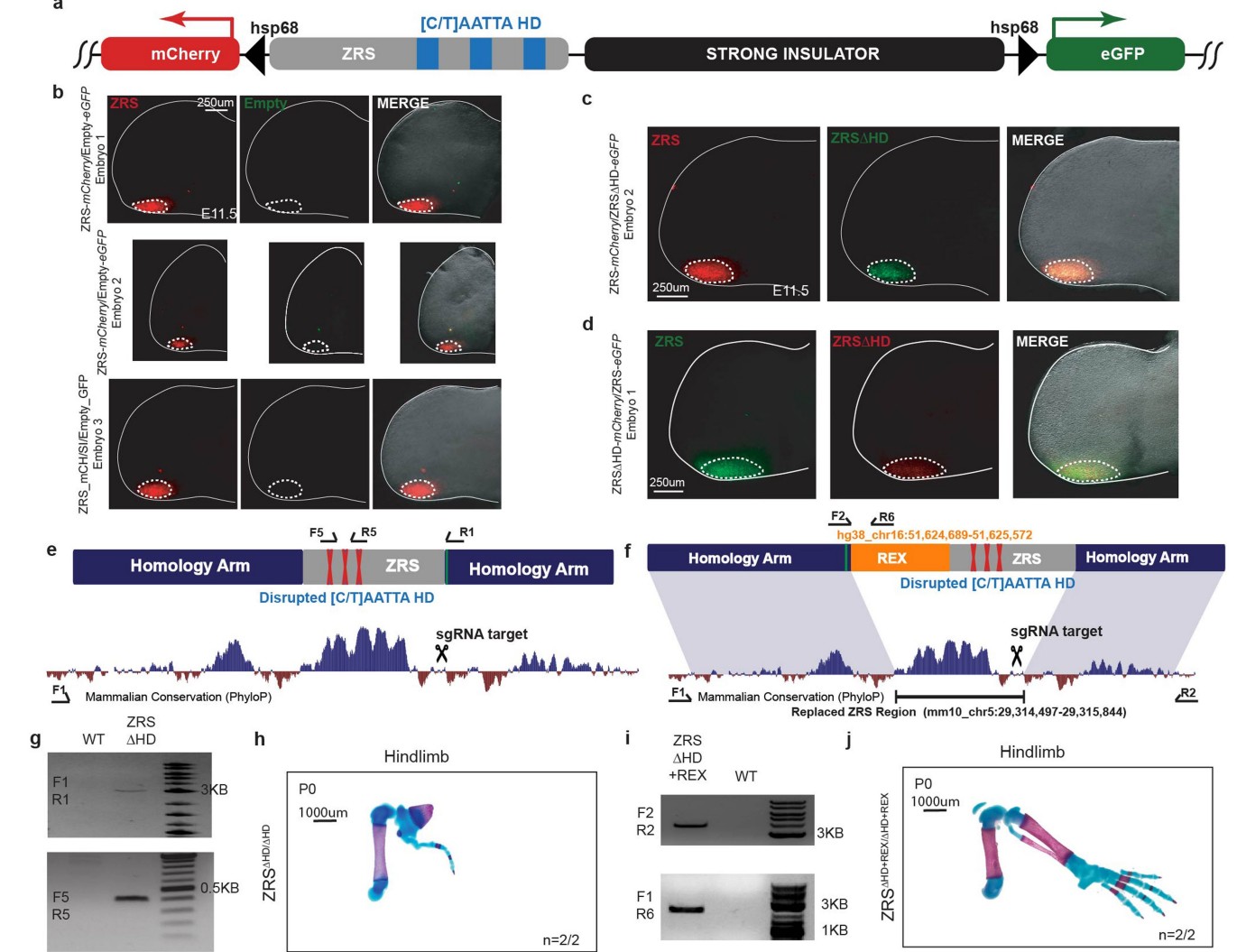

**Extended Data Fig. 7 | Mutagenesis analysis of [C/T]AATTA sites within the ZRS enhancer.** (**a**) Schematic of the ZRS-*mCherry*/Empty-*eGFP* dual-enSERT[44] vector (**b-d**) Fluorescent images of E11.5 hindlimbs of transgenic mice with a single integration of the ZRS-*mCherry*/Empty-*eGFP* dual-enSERT (**b**) ZRS-*mCherry*/ZRSΔHD-*eGFP* (n = 2) (**c**) and ZRSΔHD-*mCherry*/ZRS-*eGFP* (n = 1) (**d**) constructs at H11 locus. While dotted lines encircle the region quantified in Fig. 5. (**e-f**) Schematic of a knock-in strategy used to generate *ZRS*^ΔHD (**e**) and *ZRS*^ΔHD+REX (**f**) mice. Primers F5 and R5 are designed to anneal to the mutated

[C/T]AATTA HD sites but not the endogenous ZRS sequence. (**g**) Genotyping results of *ZRS*^ΔHD knock-in mice. (**h**) hindlimb skeletal preparations of ZRS^ΔHD/ΔHD. (**i**) Genotyping results of *ZRS*^ΔHD+REX knock-in mice. (**j**) Hindlimb skeletal preparations from *ZRS*^ΔHD+REX/ΔHD+REX P0 mice. For the PCR results shown in (**g, i**) one representative sample is shown for each genotype consistent with the results seen for all embryos and mice of the lines represented that were analysed in this study. For gel source data, see Supplementary Fig. 1.

# Reporting Summary

## Statistics

For all statistical analyses, confirm that the following items are present in the figure legend, table legend, main text, or Methods section.

| n/a | Confirmed | |
|---|---|---|
| ☐ | ☒ | The exact sample size (*n*) for each experimental group/condition, given as a discrete number and unit of measurement |
| ☐ | ☒ | A statement on whether measurements were taken from distinct samples or whether the same sample was measured repeatedly |
| ☐ | ☒ | The statistical test(s) used AND whether they are one- or two-sided *Only common tests should be described solely by name; describe more complex techniques in the Methods section.* |
| ☒ | ☐ | A description of all covariates tested |
| ☐ | ☒ | A description of any assumptions or corrections, such as tests of normality and adjustment for multiple comparisons |
| ☐ | ☒ | A full description of the statistical parameters including central tendency (e.g. means) or other basic estimates (e.g. regression coefficient) AND variation (e.g. standard deviation) or associated estimates of uncertainty (e.g. confidence intervals) |
| ☐ | ☒ | For null hypothesis testing, the test statistic (e.g. *F*, *t*, *r*) with confidence intervals, effect sizes, degrees of freedom and *P* value noted *Give P values as exact values whenever suitable.* |
| ☒ | ☐ | For Bayesian analysis, information on the choice of priors and Markov chain Monte Carlo settings |
| ☒ | ☐ | For hierarchical and complex designs, identification of the appropriate level for tests and full reporting of outcomes |
| ☒ | ☐ | Estimates of effect sizes (e.g. Cohen's *d*, Pearson's *r*), indicating how they were calculated |

*Our web collection on statistics for biologists contains articles on many of the points above.*

## Software and code

Policy information about availability of computer code

| Data collection | Sequencing data were collected on the Illumina NovaSeq 6000. Imaging data were captured through the Zeiss BioLite software (Zen Blue 3.2). qPCR data was collected using a C1000 Touch Thermal Cycler. |
|---|---|
| Data analysis | Custom algorithms or software was not developed for this research. Fluorescent images were analyzed with Fiji/ImageJ (2.14.0/1.54f).  Data were processed using trim_galore (v0.6.4), bedtools (v2.31.1), CellRanger ARC (v2.0.2), and HOMER (v4.11). Data analysis was primarily performed in R (v. >4.1.2) using a variety of published packages: Seurat (v4.4.0), Signac (v1.11.0), GenomicScores (v3.19), rstatix (v0.7.2). qPCR data was processed using CFX Maestro Software (v2.3 Bio-Rad) to extract cycle threshold values. |

For manuscripts utilizing custom algorithms or software that are central to the research but not yet described in published literature, software must be made available to editors and reviewers. We strongly encourage code deposition in a community repository (e.g. GitHub). See the Nature Portfolio guidelines for submitting code & software for further information.

## Data

Policy information about availability of data

All manuscripts must include a data availability statement. This statement should provide the following information, where applicable:
- Accession codes, unique identifiers, or web links for publicly available datasets
- A description of any restrictions on data availability
- For clinical datasets or third party data, please ensure that the statement adheres to our policy

All sequencing data generated in this study is available at the NCBI Gene Expression Omnibus (GEO) under accession number GSE243635. Capture Hi-C data from a previous study is available at GEO: GSE217078. The mm10 genome assembly and PhyloP are available at UCSC genome browser (https://hgdownload.cse.ucsc.edu/goldenpath/mm10/).

## Research involving human participants, their data, or biological material

Policy information about studies with human participants or human data. See also policy information about sex, gender (identity/presentation), and sexual orientation and race, ethnicity and racism.

| | |
|---|---|
| Reporting on sex and gender | n/a |
| Reporting on race, ethnicity, or other socially relevant groupings | n/a |
| Population characteristics | n/a |
| Recruitment | n/a |
| Ethics oversight | n/a |

Note that full information on the approval of the study protocol must also be provided in the manuscript.

# Field-specific reporting

Please select the one below that is the best fit for your research. If you are not sure, read the appropriate sections before making your selection.

☒ Life sciences          ☐ Behavioural & social sciences          ☐ Ecological, evolutionary & environmental sciences

For a reference copy of the document with all sections, see nature.com/documents/nr-reporting-summary-flat.pdf

# Life sciences study design

All studies must disclose on these points even when the disclosure is negative.

| | |
|---|---|
| Sample size | No prior analyses were used to determine the sample size before the experiment. Embryos litters were collected until at least 2 embryos of the desired genotype were acquired to show that a phenotype was reproducible. For experiments where statisical analysis was conducted (flurorescent imaging and qPCR) embryos were collected until a minimum of three replicates for each group was reached to allow for comparisons. |
| Data exclusions | Any embryos that were not at the correct developmental stage were excluded from data collection |
| Replication | ATAC-seq, ISH, skeletal staining, and florescent imaging experiments were completed with at least two biological replicates. For qPCR at least three biological replicates per each comparison group were collected. For the scATAC-seq/scRNA-seq analysis we used one biological replicate. |
| Randomization | For ATAC-seq, ISH, and skeletal staining experiments wild-type and knockin littermates were identified by numbers with genotype unknown to the investigator during data collection and sample processing. Group assignment was defined based on the mouse genotype so randomized allocation was not necessary. |
| Blinding | For ISH experiments in knockin embryos, investigators were blinded to animals' genotypes during tissue collection and in situ hybridization. For skeletal staining investigators were blinded to animals' genotypes during embryo/neonate collection and processing. For fluorescent imaging, investigators were blinded to embryos genotypes during imaging. For sequencing experiments, motif enrichment analysis, and qPCR blinding was not performed because all metrics were derived from absolute quantitative measurements without human subjectivity. |

# Reporting for specific materials, systems and methods

We require information from authors about some types of materials, experimental systems and methods used in many studies. Here, indicate whether each material, system or method listed is relevant to your study. If you are not sure if a list item applies to your research, read the appropriate section before selecting a response.

## Materials & experimental systems

| n/a | Involved in the study |
|-----|----------------------|
| ☐ | ☒ Antibodies |
| ☒ | ☐ Eukaryotic cell lines |
| ☒ | ☐ Palaeontology and archaeology |
| ☐ | ☒ Animals and other organisms |
| ☒ | ☐ Clinical data |
| ☒ | ☐ Dual use research of concern |
| ☒ | ☐ Plants |

## Methods

| n/a | Involved in the study |
|-----|----------------------|
| ☒ | ☐ ChIP-seq |
| ☒ | ☐ Flow cytometry |
| ☒ | ☐ MRI-based neuroimaging |

# Antibodies

| | |
|---|---|
| Antibodies used | Sheep Anti-Digoxigenin Fab fragments Antibody, AP Conjugated (www.sigmaaldrich.com/US/en/product/roche/11093274910) Supplier: Roche Cat# 11093274910; Lot No. 54732420 |
| Validation | Online databases: www.antibodyregistry.org/AB_514497 |

This is not a monoclonal, but a polyclonal Ab made in sheep, so there is no clone name.
After immunization with digoxigenin, sheep IgG was purified by ion-exchange chromatography, and the specific IgG was isolated by immunosorption. The Fab fragments obtained by papain digestion were purified by gel filtration, conjugated to the specific label, and stabilized in buffer. The polyclonal antibody from sheep is specific to digoxigenin and digoxin and shows no cross-reactivity with other steroids, such as human estrogens and androgens. Heat inactivation: yes

• Cross reactivity to digitoxin and digitoxigenin: <1 %
• No cross reactivity with other human estrogen or androgen steroids, e.g. estradiol or testosterone
• Cross reactivity with digoxin: not known
• Conjugate does not bind to itself at all
• Normally one molecule of the conjugate binds to one molecule digoxigenin, although ther are two possible binding sites for digoxigenin
• Nonspecific binding to RNA is not expected

More info: www.sigmaaldrich.com/deepweb/assets/sigmaaldrich/product/documents/329/822/11093274910.pdf

Here is a list of >900 citations in the manufacturer's website:
www.sigmaaldrich.com/US/en...search

# Animals and other research organisms

Policy information about studies involving animals; ARRIVE guidelines recommended for reporting animal research, and Sex and Gender in Research

| | |
|---|---|
| Laboratory animals | All animals used in this study were of Mus musculus species and FVB/NCrl strain at ages E10.5, E11.5, E13.5, E18.5, and P0. We followed standard housing specifications as outlined in IACUC Policy on Animal Housing & Environmental Enrichment: 12-hour light-dark cycle, 20-26C, humidity between 30-70%. |
| Wild animals | Study did not use wild animals. |
| Reporting on sex | Gender was not identified during embryo collections; it is assumed that the groups contained approximately equal numbers of male and female mice |
| Field-collected samples | Study did not use field-collected samples. |
| Ethics oversight | All animal work was reviewed and approved by the Lawrence Berkeley National Laboratory Animal Welfare and Research Committee and the University California Irvine Laboratory Animal Resources (ULAR). |

Note that full information on the approval of the study protocol must also be provided in the manuscript.

## Plants

Seed stocks

N/A

Novel plant genotypes

N/A

Authentication

N/A

