## [Peer Review File · Nature]

Range Extender Mediates Long-Distance Enhancer Activity

Corresponding Author: Dr Evgeny Kvon

This file contains all reviewer reports in order by version, followed by all author rebuttals in order by version. Parts of this Peer Review File have been redacted as indicated to remove third-party material.

Version 0:

Reviewer comments:

Referee #1

(Remarks to the Author)

This paper by Grace Bower et al. from the Kvon lab addresses an important and highly relevant question. They investigate in vivo how long-range enhancers communicate with their target genes. Most of this process is unknown although a number of investigations point to factors including 3D chromatin folding, CTCF binding, and the existence of elements other than bona fide enhancers such as so-called booster elements or enablers (e.g. Blayney JW Cell 2023) that are necessary for function but do not have enhancer activity by themselves. Bower et al. claim to have identified an element that is necessary as well as sufficient to confer long range enhancer activity. They investigate this at a locus that has been studied extensively before, also by the last author, the long range Shh enhancer ZRS. They show that other enhancers active in the limb cannot replace the ZRS in spite of the fact that their native regulatory activity overlaps with that of the ZRS. An additional element they call REX is identified that apparently is capable of transferring this long-range activity. In an elegant series of experiments, they show that this REX element has LHX binding sites and that these are present in many long-range enhancer. Mutation of these sites in the ZRS destroys its long-range activity but not its short-range ability to activate Shh.

Overall, this is a nice piece of work with a convincing set of experiments. What needs to be considered, however, is the very special nature of the Shh regulatory landscape and its enhancer ZRS. The ZRS is one of the few examples of a non-redundant enhancer regulating a developmentally essential gene. This situation is rather exceptional and thus has to be considered when drawing general conclusions. Furthermore, the ZRS is embedded in CTCF sites that tether it to the Shh promoter and are necessary for normal function (Ushiki A Nat Commun. 2021; Paliou, PNAS 2019).

Another point to consider are local-specific effects and the conclusions drawn from transgenic experiments in which enhancers are placed right next to promoters. What is usually used, and that is also the case here, are minimal promoters that apparently confer little specificity. But this may be due to their small size and the more or less arbitrary cut off from other flanking sequence. Thus, a clean experiment should involve insertion of the enhancer upstream at a distance of a few kb from promoter histone marks and test its ability to regulate and activate the target gene in its natural environment. Thus, the conclusion that the tested enhancers are capable of short-range activation of Shh is limited to this experimental setting and cannot be generalized.

Another general concern is that the entire ms is directed towards long-range regulation but no long-range contacts are shown. Again, the ZRS is somewhat special as the enhancer is apparently always close the promoter. Shh/ZRS co-localization correlates with the spatiotemporal domain of limb bud-specific Shh expression, but also occurs regardless of whether the gene or enhancer is active (Williamson, Development 2016). This makes conclusions about long-range activity difficult as this is unlikely to be the case for most other loci. In fact, it has been reported that some genes contact their enhancers via pre-formed complexes whereas other are in contact only when being activated.

The authors identify a region they call REX that they claim confers long range activation of the target gene. They include this sequence in the transplanted enhancer and can demonstrate Shh activity. This is a beautiful experiment but it basically shows that the enhancer needs an enabling element to function. For the argument that it works at short distance – see above. What the authors should do is create a knock out of this element and show that the enhancer fails to interact/activate Sall1 at its native locus.

Another important question is if the enhancer/REX or the nearby sequence has CTCF binding in the limb. Such a site may confer increased looping and long-range activity.

A somewhat puzzling observation is the rather specific ZRS expression of the transplanted enhancer. Why would this enhancer that normally has an activity throughout the limb become restricted to the ZRS?

The identification of Lhx and Lef binding sites is interesting and points towards a possible mechanism. Nevertheless, if long-range enhancers get in contact with their cognate promoters via binding to such TFs, one wonders how they might achieve their specificity. Long range enhancers apparently do not contact other neighboring genes even if the TAD environment is lost. If long-range enhancers would act on any promoter activate in the same tissue as suggested here, one would expect

wide spread activation, which is not the case. Thus, I would urge the authors to reduce their claim that this is a general mechanism explaining long-range enhancer activity.

Finally, the authors inactivate Lhx binding sites in the ZRS and observe a loss of activity with a typical phenotype. If the REX element has, as they claim, a universal function, one would expect that the mutant can be rescued by adding the REX element to the ZRS. This important experiment would show that the sequence is indeed sufficient to induce long range regulation.

Referee #2

(Remarks to the Author)

The manuscript by Bower et al, addresses a very current, and long-standing question – how do distal enhancers acting over very long-range regulate their target genes?

The authors first show through a series of elegant genetic experiments in mice that short and medium range enhancers cannot function over long-distances. They then discover a sequence element that is located beside a long-range enhancers and not short-range enhancers. This element is not an enhancer itself, but facilitates long range enhancers to function. It is even sufficient to confer long-range activity to an enhancer that normally can only act over shorter distances.

They confirmed that the reason short-range enhancers could not work over a distance – when placed at 840kb away was not due to (i) inaccessible – the elements were open, (ii) promoter incompatibility (by using the Shh promoter in a transgene). They identify a homeobox motif that is globally enriched in long distance limb enhancers and not in short distant enhancers. Mutation of this site is not essential for enhancer activity, but is essential for the ZRS long distance enhancer activity

Overall this is a great body of work, that will be of broad interest to a wide-spectrum of readers. I have two suggestions (minor) that will improve the paper.

1)) A key dataset to find the homeobox motif was the development of a high-confidence list of long-range and short-range limb enhancers. Assigning enhancers to their target genes is a current huge challenge, as the authors know well – especially over distances out to 2Mb. The description of how this was done is not described in enough detail – lines 940-947.

The methods says that the Multiome E-P assignments or the enhancer Capture HiC was used. Do you really mean 'or' here? If yes, which was given preference if they both gave different genes? The multiome assignment will give many false pairs – especially going out to 2Mb. It would make more sense to use 'and' – i.e. to filter those based on which ones also had a capture HiC interaction.

Statistically, I can understand why you did the following “Only the longest-distance gene was considered for enhancers with multiple target genes to prevent statistical overrepresentation.” – but this is likely enriched for false positive E-P pairs. Did you check the spatial expression of any of these genes? Obviously – globally you had enough signal to find the motif, which is the main thing, but it would still be very useful to explain this in more detail and to also present the caveats. It is likely that you have a lot of noise (false pairing) – which is perhaps why the difference in the E-P distances is not that impressive for enhancer with none or 1 HD motif versus >1 (Supple Fig. 6).

Line 948 – in relation to the motif analysis the sentence “E-P sets for (1) the bona fide E-P pairs (using VISTA enhancer coordinates)”. Here you should change the wording – they are still predicted E-P pairs, but for bona fide enhancers (the pairing is a prediction).

The multipome based pairs – are predictions for both the putative enhancer and the pairing to a gene

2) I suggest that the authors carefully reconsider calling these regions with a new name – REX. I don't agree with the authors discussion trying to distinguish these from tethering elements discovered in Drosophila (work from Mike Levine's lab). Conceptually, these seem to be the very similar if not the same. There is enough renaming within the field without adding to this. It dilutes the literature, confuses students and dilutes your findings.

Having a conserved process – requiring an additional element placed beside or embedded within an enhancer that is specifically involved in proximity/looping of long-range elements rather than enhancer activity is similar to tethering elements and PRE binding elements within or beside enhancers.

Referee #3

(Remarks to the Author)

The manuscript by Bower et al. uses a series of elegant genetic experiments to demonstrate existence of the novel 'Range extender element' (REX). Although lacking enhancer activity itself, REX confers ability of enhancers to activate expression over extreme long ranges. The authors further show that REX element and other long-range limb enhancers contain highly conserved [C/T]AATTA homeodomain (HD) motifs. Mutation of these motifs in a canonical extreme long-range enhancer, ZRS, results in limb malformations typical of the loss Shh expression. Overall, this is a well-executed, written and presented study. The findings are novel and for the first time implicate HD motif-containing elements in extending the range of long-range gene regulation. These observations are of broad interest to transcriptional regulation, genome organization and developmental biology fields. I only have a few comments that should be addressed before publication.

Major comments:

1. The authors claim that disruption of [C/T]AATTA motifs does not change ZRS enhancer activity in a reporter assay, but I am not fully convinced this is indeed the case. The HD motif mutant enhancer does look weaker (Fig. 6C), though perhaps the mean intensity from mCH and GFP cannot be directly compared. How would the comparison look if the mutant and wild type ZRS enhancers were swapped in this reporter? The possibility that the HD motif mutations do quantitatively affect ZRS enhancer activity should be at least acknowledged, and it may confound the author's interpretation of the results in Fig. 6D. See also next point, which would be a more direct way to accomplish a separation of function.

2. Does mutation of the three HD motif sites within the REX element (Fig. 5D) kill its ability to rescue Shh expression and limb formation, when coupled to HS72? This is an important experiment, since in this setup, the extender and enhancer functions can be more easily genetically separated.

3. Beyond the importance of the HD motifs, a current study provides little insight into the mechanism by which extender elements might work. While detailed molecular investigation is – in my view at least – beyond the scope of the current study, a couple of straightforward genomic analyses may provide some additional insights. For example, the [C/T]AATTA motif is bound by many HD motif TFs expressed in the developing limb. Based on the available ChIP-seq data, is there any information on which of these HD TFs may be preferentially binding at the long-range vs short-range hindlimb enhancers (using extended regions, as in Fig. 5)? Is there an enrichment of cohesin at the long-range enhancers compared to the short-range enhancers? Presumably, the REX element is CTCF independent. Can authors confirm that it is not bound by CTCF in the limb?

Version 1:

Reviewer comments:

Referee #1

(Remarks to the Author)

The authors performed an impressive set of additional experiments with several new transgenic mice. These results provide important additional evidence for the claim of the paper. The results are convincing. The authors have also toned down the message of the ms and included relevant limitations of their study in the discussion and in other parts of the ms.

Referee #2

(Remarks to the Author)

The authors have addressed all of my concerns in full (I will leave the naming issue up to the authors).

The two additional knock in lines with the ZRS replaced by the mutant HS72 enhancer (both the LEF1 motif alone, and the LHX motifs) is a great addition that really strengthens the manuscript.

Overall, this is a really lovely body of work that the authors can be very proud of. It will become a seminal paper in the field.

Referee #3

(Remarks to the Author)

I am completely satisfied with the revision and want to congratulate the authors on their elegant and important study.

Point-by-Point Response to Reviewers' Comments

Nature resubmission #2024-05-09393A Bower et al.: "Conserved Cis-Acting Range Extender Element Mediates Extreme Long-Range Enhancer Activity in Mammals"

Overview

We thank the reviewers for their positive response and for recognizing the significance of our findings for gene regulation, genome organization and developmental biology. We are also grateful for the reviewers' rigorous feedback and suggestions on how to improve the manuscript. We have substantially revised the manuscript, working to address all reviewers' comments in order to strengthen it for publication in *Nature*. As part of the revision experiments, we generated and phenotyped five new knock-in mouse lines, all of which reinforce the main conclusions of the paper. Major revisions are: 1) We showed that integration of the heterologous HS72 enhancer upstream of the endogenous *Shh* gene results in an increase of *Shh* expression in the limb confirming that HS72 enhancer is capable of short-range *Shh* activation in *Shh*'s natural genomic environment; 2) We mutagenized LHX sites within the REX element itself and show through knock-in experiments at the *Shh* locus that they are critical for its long-range activity; 3) We show through further knock-in experiments at the *Shh* locus that long-range activity of the ZRS enhancer lacking LHX sites can be rescued by adding REX, directly demonstrating that the LHX sites found in the REX element and the ZRS are functionally equivalent; 4) We toned down the claim about the generality of the discovered mechanism and revised the TF motif analysis. Please see our point-by-point response to the reviewers' comments below.

Reviewer 1

Reviewer 1, Overview: This paper by Grace Bower et al. from the Kvon lab addresses an important and highly relevant question. They investigate in vivo how long-range enhancers communicate with their target genes. Most of this process is unknown although a number of investigations point to factors including 3D chromatin folding, CTCF binding, and the existence of elements other than bona fide enhancers such as so-called booster elements or enablers (e.g. Blayney JW Cell 2023) that are necessary for function but do not have enhancer activity by themselves. Bower et al. claim to have identified an element that is necessary as well as sufficient to confer long range enhancer activity. They investigate this at a locus that has been studied extensively before, also by the last author, the long range *Shh* enhancer ZRS. They show that other enhancers active in the limb cannot replace the ZRS in spite of the fact that their native regulatory activity overlaps with that of the ZRS. An additional element they call REX is identified that apparently is capable of transferring this long-range activity. In an elegant series of experiments, they show that this REX element has LHX binding sites and that these are present in many long-range enhancer. Mutation of these sites in the ZRS destroys its long-range activity but not its short-range ability to activate *Shh*.

Reviewer 1, Specific Point 1a: Overall, this is a nice piece of work with a convincing set of experiments. What needs to be considered, however, is the very special nature of the *Shh* regulatory landscape and its enhancer ZRS. The ZRS is one of the few examples of a non-

redundant enhancer regulating a developmentally essential gene. This situation is rather exceptional and thus has to be considered when drawing general conclusions.

We thank the reviewer for summarizing our work, putting it in the broader context of other studies, and highlighting the significance of our findings. The ZRS presents a unique opportunity to study these types of mechanisms because of the lack of redundant enhancers and the clear and well-studied phenotypes resulting from ZRS mutations. The long-range mechanism that we identified would have been difficult to decipher using enhancer replacement experiments at other enhancer loci because of the confounding effects of other redundant enhancers. That said, we fully agree with the reviewer that the ZRS, in part because of these features, represents an unusual case compared to most other loci and the known peculiarities of this locus need to be carefully taken into account in the interpretation of our results.

We have now performed the rescue experiment suggested by the reviewer, showing that the addition of REX to the ZRS enhancer with mutated LHX sites rescues its function (see the response to **Specific Point 8** and new **Fig. 6e**). We have also shown that we can combine REX from one enhancer with another short-range enhancer (2nd enhancer locus) and insert this combination in place of the ZRS (3rd enhancer locus) and this three-locus configuration results in long-range activation (**Fig. 3d**). We have also demonstrated strong enrichment of [C/T]AATTA in long-range enhancers genome-wide (**Fig. 5** and **Extended Data Fig. 6**). In combination, we believe that these three complementary lines of evidence provide strong support for the notion that the observed sequence signature and its involvement in long-range interaction is not specific to the *Shh* locus but part of a more common mechanism enabling long-range interactions.

That said, we acknowledge the limitations of our enhancer replacement experiments and revised the abstract, results and discussion accordingly to more explicitly highlight the limitations of the study and that the experiments were performed at *Shh* locus and only in limb bud tissue. We also added the following new paragraph to the discussion in the revised manuscript:

Discussion (Pg 22 Ln 1-5): Because our enhancer substitution experiments were performed at the *Shh* locus, we cannot rule out the potential impact of locus-specific effects on our findings. Further investigation into long- versus short-range enhancer activity at other genomic loci and across different cell types will be necessary to understand how widespread REX elements are in mammalian genomes.

Reviewer 1, Specific Point 1b: Furthermore, the ZRS is embedded in CTCF sites that tether it to the *Shh* promoter and are necessary for normal function (Ushiki A Nat Commun. 2021; Paliou, PNAS 2019).

We agree that CTCF contributes to the ZRS long-range activation of *Shh* in the context of limb development, despite being dispensable for reactivation of the ZRS in mESCs (Kane et al. 2022). As the reviewer pointed out, the *in vivo* mouse studies by Paliou et al 2019 clearly show that CTCF sites are necessary for *Shh* activation. However, there must be other factors that are important for long-range activation at *Shh* and other loci since deletion of both CTCF sites

surrounding the ZRS in mice results in normal limbs and *Shh* expression is only reduced by 50% (Paliou PNAS 2019). Similarly, at other long-range loci, deletion of enhancer- and promoter-proximal CTCF sites only mildly affects gene expression (e.g., *Sox9*; Chen et al 2023). In our view, the [C/T]AATTA sequence signature is another genetic mechanism, likely independent of CTCF, that contributes to long-range activation at the *Shh* locus, as we highlight in the introduction. We thank the reviewer for bringing up this important point.

Reviewer 1, Specific Point 2: Another point to consider are local-specific effects and the conclusions drawn from transgenic experiments in which enhancers are placed right next to promoters. What is usually used, and that is also the case here, are minimal promoters that apparently confer little specificity. But this may be due to their small size and the more or less arbitrary cut off from other flanking sequence. Thus, a clean experiment should involve insertion of the enhancer upstream at a distance of a few kb from promoter histone marks and test its ability to regulate and activate the target gene in its natural environment. Thus, the conclusion that the tested enhancers are capable of short-range activation of *Shh* is limited to this experimental setting and cannot be generalized.

We agree with the reviewer about the limitations of transgenic reporter assays. It is indeed possible that an enhancer can activate *Shh* in the context of the transgene but not at the endogenous locus where native genomic context can impact the short-range activation. We followed the reviewer's suggestion and generated a new knock-in line in which the HS72 enhancer without REX element was inserted ~2 KB upstream of the *Shh* endogenous promoter. The resulting knock-in mice had polydactylous limbs and elevated *Shh* expression in both posterior and anterior limb buds, providing direct evidence that the HS72 enhancer is capable of short-range *Shh* activation at the native *Shh* locus. We thank the reviewer for suggesting this important control experiment. We added these new results on **Pg 10 Ln 9-24** and as new **Fig. 2c** and **Extended Data Fig. 4** in the revised manuscript:

Results (Pg 10 Ln 9-24): To test if the HS72 enhancer can activate gene expression at the endogenous *Shh* locus we generated a knock-in line in which the HS72 enhancer sequence was inserted ~2 kb upstream of the endogenous *Shh* promoter (Fig. 2c). Preaxial polydactyly (formation of extra digits) provides a sensitive and readily accessible phenotypic readout of ectopic *Shh* misexpression beyond the ZPA of the developing limb^{5,38}. Since HS72 is active more broadly than the ZRS in limb mesenchyme, the presence of polydactyly would indicate that it can activate functional SHH signaling. Indeed, *Shh*^{HS72(+2KB)/wt} mice had polydactylous fore- and hindlimbs and *de novo Shh* expression in anterior hindlimb buds consistent with the HS72 enhancer driving expanded *Shh* activity⁵⁶ (**Fig. 2c** and **Extended Data Fig. 4b**). This result was similar to the polydactyly observed in transgenic mice in which the HS72::*Shh* transgene was integrated at a safe-harbor location in the mouse genome (**Extended Data Fig. 4a**). Importantly ZRS^{HS72/wt} mice had normal limbs indicating absence of broad HS72-mediated *Shh* activation from a remote

position (**Extended Data Fig. 1e**). Taken together, these results show that the HS72 enhancer can activate *Shh* promoter at short range in its native endogenous context and in the context of a transgene.

Reviewer 1, Specific Point 3: Another general concern is that the entire ms is directed towards long-range regulation but no long-range contacts are shown. Again, the ZRS is somewhat special as the enhancer is apparently always close the promoter. *Shh*/ZRS co-localization correlates with the spatiotemporal domain of limb bud-specific *Shh* expression, but also occurs regardless of whether the gene or enhancer is active (Williamson, Development 2016). This makes conclusions about long-range activity difficult as this is unlikely to be the case for most other loci. In fact, it has been reported that some genes contact their enhancers via pre-formed complexes whereas other are in contact only when being activated.

We agree with the reviewer that the ZRS/*Shh* is a special locus due to surrounding CTCF sites and the formation of “stable” structural CTCF loops, which cause *Shh*/ZRS proximity in cells where the enhancer is not active. These structural CTCF loops are likely a separate mechanism from the one we identified in the current study (see our response to **Specific Point 1b**). However, because these structural CTCF loops form even in the absence of the ZRS enhancer (Chen et al. 2024), we believe that it would be challenging to interpret the outcomes of 3C or FISH-type experiments in our knock-in mice. We also note that the conclusions of the present study are robust to either outcome of those experiments and that the name Range Extender does not imply a long-range contact mechanism, as is, for example, the case for fly tethering elements where the name implies a looping mechanism. The discovery of the REX element poses lots of questions for future studies, including the molecular mechanism, the role of looping, and whether it applies to other loci, which, in our opinion, are outside the scope of the current study (to which reviewer #3 also agrees). To acknowledge this, we revised the Discussion section accordingly (see Pg 21 Ln 7-26).

Reviewer 1, Specific Point 4: The authors identify a region they call REX that they claim confers long range activation of the target gene. They include this sequence in the transplanted enhancer and can demonstrate *Shh* activity. This is a beautiful experiment but it basically shows that the

enhancer needs an enabling element to function. For the argument that it works at short distance – see above. What the authors should do is create a knock out of this element and show that the enhancer fails to interact/activate *Sall1* at its native locus.

We agree that it would be very interesting to determine if the REX element is required for long-range enhancer activation of *Sall1*. Our previously published full deletion of HS72+REX from the mouse genome does not lead to any limb phenotypes, nor does it result in detectable changes in *Sall1* expression (by qPCR and WISH), suggesting that there are likely other redundant limb enhancers in the locus which are uncharacterized (Osterwalder et al. 2018):

FIGURE REDACTED

Since the genomic location and exact activity patterns of this redundant enhancer(s) remain elusive, we believe that it would be challenging to interpret the outcomes of REX-only deletions at its native *Sall1* locus.

However, we agree with the reviewer that additional evidence for the functional significance of the REX element and the critical sequence motifs would strengthen the study. We have now generated two additional knock-in mouse lines in which the ZRS was replaced with the extended HS72 sequence containing the REX element with either the single LEF1 motif or both LHX motifs mutagenized. Only knock-in mice with disrupted LHX motifs resulted in mice with truncated limbs, indicating that HD motifs are required for REX function. We added these new results on **Pg 14 Ln 4-12** and as new **Fig. 4** in the revised manuscript:

Results (Pg 14 Ln 4-12): To determine the importance of LHX and LEF1 motifs for REX function, we generated two knock-in mice in which the ZRS was replaced with the extended HS72 sequence containing the REX element with either the single LEF1 motif or both LHX motifs mutagenized. Mice containing the REX element with mutated LEF1 motif showed fully developed limbs with polydactyly indicating that disruption of the LEF1 motif does not abolish long-range activity of the REX element (**Fig. 4b**). However, disrupting both LHX motifs resulted in mice with truncated limbs, indicative of a loss of *Shh* expression in the limb bud (**Fig. 4c**). Together, these results demonstrate that the LHX motifs in the REX element are required for its ability to facilitate long-range enhancer activity at *Shh* locus.

Reviewer 1, Specific Point 5: Another important question is if the enhancer/REX or the nearby sequence has CTCF binding in the limb. Such a site may confer increased looping and long-range activity.

We agree that this is an important consideration. We scanned the REX element for the presence of CTCF or YY1 motifs and did not find any. We also inspected a published CTCF ChIP-seq dataset from E11.5 limb buds (Andrey et al. 2017) and determined that neither HS72 nor REX have any CTCF binding *in vivo*. We added these new results on **Pg 13 Ln 22-23** and as new **Extended Data Fig. 6e** in the revised manuscript:

Results (Pg 13 Ln 22-23): The REX element lacks CTCF and YY1 motifs and is not bound by CTCF⁵⁷ in embryonic limb buds (Extended Data Fig. 6e).

Reviewer 1, Specific Point 6: A somewhat puzzling observation is the rather specific ZRS expression of the transplanted enhancer. Why would this enhancer that normally has an activity throughout the limb become restricted to the ZRS?

We agree with the reviewer that this is an intriguing result for which we do not have an explanation. The factors that restrict *Shh* expression in knock-in mice could, for example, include regulation at the promoter level, repression coming from a genomic locus, or post-transcriptional repression. One factor that can at least partially explain the observed result is a distal limb mesenchyme-specific expression of LHX2 and LHX9 in the limb bud. However, their expression in the limb is much broader than observed *Shh* activity, so there must be other mechanisms. It will be interesting to figure this out in the future studies. We added this discussion on **Pg 21 Ln 13-17** in the revised manuscript:

Results (Pg 21 Ln 13-17): The [C/T]AATTA motifs match the binding preferences of many HD TFs, most notably LIM-homeodomain TFs LHX2 and LHX9. LHX2 and LHX9 are required for normal limb outgrowth and are expressed in distal limb mesenchyme⁷⁵ potentially restricting the activity of transplanted enhancers to the posterior and anterior portions of distal limb mesenchyme (**Extended Data Fig. 5c**).

Reviewer 1, Specific Point 7: The identification of Lhx and Lef binding sites is interesting and points towards a possible mechanism. Nevertheless, if long-range enhancers get in contact with their cognate promoters via binding to such TFs, one wonders how they might achieve their specificity. Long range enhancers apparently do not contact other neighboring genes even if the TAD environment is lost. If long-range enhancers would act on any promoter activate in the same tissue as suggested here, one would expect wide spread activation, which is not the case. Thus, I would urge the authors to reduce their claim that this is a general mechanism explaining long-range enhancer activity.

We agree with the reviewer that our findings do not explain how enhancer-promoter specificity within TADs (and in their absence) is regulated, which is one of the greatest unresolved questions in gene regulation. However, we believe that the questions of long-range activity and promoter specificity are two separate questions. While we identified sequence elements required and sufficient for long-range activity, we cannot say anything about how they regulate promoter specificity. We, therefore, revised the abstract, introduction, and discussion accordingly to more explicitly highlight the limitations of the study and highlight that the experiments were performed at *Shh* locus. We also added the following new paragraph to the discussion in the revised manuscript, highlighting that E-P specificity is still an unresolved question:

Discussion (Pg 22 Ln 1-7): Because our enhancer substitution experiments were performed at the *Shh* locus, we cannot rule out the potential impact of locus-specific effects on our findings. Further investigation into long- versus short-range enhancer activity at other genomic loci and across different cell types will be necessary to understand how widespread REX elements are in mammalian genomes. Other questions posed by our study are how the promoter specificity of long-range REX-mediated interactions is regulated and whether loop extrusion plays a role.

Reviewer 1, Specific Point 8: Finally, the authors inactivate Lhx binding sites in the ZRS and observe a loss of activity with a typical phenotype. If the REX element has, as they claim, a universal function, one would expect that the mutant can be rescued by adding the REX element to the ZRS. This important experiment would show that the sequence is indeed sufficient to induce long range regulation.

We agree that this is an important experiment to further confirm the universality of the REX element. We generated a new knock-in mouse line in which the endogenous ZRS with mutated LHX motifs was appended by the REX element as suggested by the reviewer. The resulting ZRS^{ΔHD+REX/-} knock-in mice fully developed zeugopod and part of the autopod with 3 digits on both fore- and hindlimbs, indicating partial rescue of limb outgrowth even in mice containing only one copy of the ZRS^{ΔHD+REX} allele. We thank the reviewer for suggesting this important experiment to confirm REX element universality. We added these new results on **Pg 18 Ln 10-22** and as new **Fig. 6e** and **Extended Data Fig. 7** in the revised manuscript:

Results (Pg 18 Ln 10-22): The loss of limb outgrowth in ZRS^{ΔHD/ΔHD} mice could be near-completely rescued by addition of the REX element; mice containing one copy of the [C/T]AATTA-less ZRS fused to the REX element fully developed zeugopod and part of the autopod with 3 digits on both fore- and hindlimbs (**Fig. 6e**).

Taken together, our knock-in and transgenic results indicate that [C/T]AATTA HD motifs in the ZRS are dispensable for limb-specific activity at short range, but critically required for long-range activity. These experiments uncouple the remarkable tissue specificity of the ZRS, directing highly restricted expression to the ZPA, from its ability to act over extremely long genomic distances. Our results also indicate that the loss of long-range enhancer activity upon removal of endogenous [C/T]AATTA HD motifs could be compensated by addition of a heterologous REX element, demonstrating that the [C/T]AATTA HD motifs found in the REX element and the ZRS are functionally equivalent.

Reviewer 2

Reviewer 2, Overview: The manuscript by Bower et al, addresses a very current, and long-standing question – how do distal enhancers acting over very long-range regulate their target genes?

The authors first show through a series of elegant genetic experiments in mice that short and medium range enhancers cannot function over long-distances. They then discover a sequence element that is located beside a long-range enhancers and not short-range enhancers. This element is not an enhancer itself, but facilitates long range enhancers to function. It is even sufficient to confer long-range activity to an enhancer that normally can only act over shorter distances.

They confirmed that the reason short-range enhancers could not work over a distance – when placed at 840kb away was not due to (i) inaccessible – the elements were open, (ii) promoter incompatibility (by using the Shh promoter in a transgene). They identify a homeobox motif that is globally enriched in long distance limb enhancers and not in short distant enhancers. Mutation of this site is not essential for enhancer activity, but is essential for the ZRS long distance enhancer activity

Overall this is a great body of work, that will be of broad interest to a wide-spectrum of readers.

We thank the reviewer for their positive feedback and for recognizing the broad significance of our findings.

Reviewer 2, Specific Point 1a: A key dataset to find the homeobox motif was the development of a high-confidence list of long-range and short-range limb enhancers. Assigning enhancers to their target genes is a current huge challenge, as the authors know well – especially over distances out to 2Mb. The description of how this was done is not described in enough detail – lines 940-947.

We agree with the reviewer that E–P assignments based on Capture Hi-C and especially multiome analysis are noisy. To reduce the number of false positive E-P pairs, we only considered genes that are expressed in limb mesenchyme based on published RNA-seq and our scRNA-seq datasets. We also performed new analyses suggested by the reviewer to reduce the false positive rate among E-P pairs. These new analyses, which we included in the revised manuscript, confirm the enrichment of LHX motifs in long-range enhancers and are described in detail below. We also revised the **Methods** and **Fig. 5** legends to clarify the details of the E-P assignment and TF motif analysis.

Reviewer 2, Specific Point 1b: The methods says that the Multiome E-P assignments or the enhancer Capture HiC was used. Do you really mean ‘or’ here? If yes, which was given preference if they both gave different genes?

Yes, it is “or”. We used both datasets to increase the number of long- and short-range VISTA limb enhancers. If both methods called different genes for the same enhancer, we considered both as possible targets. We clarified it in the **Methods** of the revised manuscript.

Reviewer 2, Specific Point 1c: The multiome assignment will give many false pairs – especially going out to 2Mb. It would make more sense to use ‘and’ – i.e. to filter those based on which ones also had a capture HiC interaction.

Following the reviewer's suggestion, we repeated the motif analyses for Capture-Hi-C-called VISTA E-P interactions as well as for VISTA E-P interactions called by Capture-Hi-C and multiome methods (**Extended Data Fig. 6c-h**). This filtering resulted in much fewer long- and short-range enhancers, but we observed similar LHX-motif enrichment trends:

Reviewer 2, Specific Point 1d: Statistically, I can understand why you did the following “Only the longest-distance gene was considered for enhancers with multiple target genes to prevent statistical overrepresentation.” – but this is likely enriched for false positive E-P pairs. Did you check the spatial expression of any of these genes? Obviously – globally you had enough signal to find the motif, which is the main thing, but it would still be very useful to explain this in more detail and to also present the caveats. It is likely that you have a lot of noise (false pairing) – which is perhaps why the difference in the E-P distances is not that impressive for enhancer with none or 1 HD motif versus >1 (Supple Fig. 6).

We only considered genes that are expressed in limb mesenchyme based on published RNA-seq and our scRNA-seq datasets. We also agree with the reviewer that the longest EP interactions are likely enriched for false positive interactions. We, therefore, repeated all analyses using the closest predicted target gene for each enhancer. Notably, the difference in E-P distance between enhancers with 0-1 and >1 LHX motifs became even more pronounced for VISTA enhancers but

not for those predicted by scATAC-seq enhancers, which showed a reverse trend, possibly due to a high fraction of false-positive enhancers identified by scATAC-seq.

We thank the reviewer for these helpful suggestions. We included all the above analyses as part of the revised **Fig. 5** and **Extended Data Fig. 6**. We also revised the **Methods** to clarify the details of the E-P assignment and TF motif analysis.

Reviewer 2, Specific Point 2: Line 948 – in relation to the motif analysis the sentence “E-P sets for (1) the bona fide E-P pairs (using VISTA enhancer coordinates)”. Here you should change the wording – they are still predicted E-P pairs, but for bona fide enhancers (the pairing is a prediction).

The multipome based pairs – are predictions for both the putative enhancer and the pairing to a gene

We agree that the phrasing is confusing, and have revised the methods section accordingly:

Methods (Pg 32, 33 Ln 28-35, 1-6): A list of functionally-validated limb enhancers was obtained from the VISTA browser dataset⁵⁸. To assign putative target genes for these enhancers, we used the multiome E-P assignments or enhancer capture Hi-C data 12. Target genes were filtered to limb-expressed genes (curated from the hindlimb scRNA-seq (this study) and whole-limb bulk RNA-seq (Limb-Enhancer Genie⁹⁴) were considered in the analysis. For enhancers with multiple target genes we considered the longest (or shortest, see **Extended Data Fig. 6**) distance interaction followed by separation into two groups based on the distance between the enhancer and target gene: short-range (10 - 200 kb) and long-range (400 kb - 2 Mb). We then performed differential motif enrichment analysis using the findMotifsGenome.pl command in HOMER with a given size⁹⁵ comparing short- and long-range E-P sets for (1) predicted E-P pairs for bona fide enhancers (using VISTA enhancer coordinates) and (2) predicted E-P pairs for putative limb enhancers defined by scATAC-seq. HOMER and JASPAR2022⁹⁶ motifs were

utilized in the motif search. Only motifs for limb-expressed TFs, from the same list used to filter target genes, were considered in the analysis.

We also updated the figure legend to better clarify this as well:

(Pg 16 Ln 2-6): Table of top-most enriched TF motifs in long-range (400 kb - 2 Mb), as compared to short-range (10-200 kb) enhancers for experimentally verified VISTA limb enhancers assigned to target genes by single-cell ATAC-seq/RNA-seq or Capture Hi-C and for predicted hindlimb enhancers assigned to target genes by single-cell ATAC-seq/RNA-seq. Only the farthest target gene (within 2 Mb range) was considered for each enhancer.

Reviewer 2, Specific Point 3: I suggest that the authors carefully reconsider calling these regions with a new name – REX. I don't agree with the authors discussion trying to distinguish these from tethering elements discovered in Drosophila (work from Mike Levine's lab). Conceptually, these seem to be the very similar if not the same. There is enough renaming within the field without adding to this. It dilutes the literature, confuses students and dilutes your findings.

Having a conserved process – requiring an additional element placed beside or embedded within an enhancer that is specifically involved in proximity/looping of long-range elements rather than enhancer activity is similar to tethering elements and PRE binding elements within or beside enhancers.

We agree with the reviewer that the identified REX element is conceptually similar to the tethering elements described by Mike Levine and others in flies. We also agree that the unnecessary propagation of new terms for genetic elements of similar functions is unnecessary and creates confusion, especially for people outside the field. The reason we did not use the term “tethering element” is because this name implies a direct tethering mechanism between two elements, the existence of which was convincingly shown in flies through genetic, imaging and HiC experiments. In the case of the REX element, the molecular mechanism remains to be established and will be a subject of future studies. It is also likely that the mechanisms of action of fly tethering elements and REX elements are different. Well-characterized fly tethering elements work through GAGA factor binding to GC-rich motifs that do not have known homologs in mammals. We, therefore, called these elements based on their ability to extend the range of the enhancer and not based on the tethering mechanism for which there is no evidence. We hope that the reviewer agrees to this more factual nomenclature.

Reviewer 3

Reviewer 3, Overview: The manuscript by Bower et al. uses a series of elegant genetic experiments to demonstrate existence of the novel 'Range extender element' (REX). Although lacking enhancer activity itself, REX confers ability of enhancers to activate expression over extreme long ranges. The authors further show that REX element and other long-range limb enhancers contain highly conserved [C/T]AATTA homeodomain (HD) motifs. Mutation of these motifs in a canonical extreme long-range enhancer, ZRS, results in limb malformations typical of the loss Shh expression. Overall, this is a well-executed, written and presented study. The findings are novel and for the first time implicate HD motif-containing elements in extending the range of long-range gene regulation. These observations are of broad interest to transcriptional regulation, genome organization and developmental biology fields. I only have a few comments that should be addressed before publication.

We are grateful for the reviewer's positive feedback and recognition of the impact of our work.

Reviewer 3, Specific Point 1: The authors claim that disruption of [C/T]AATTA motifs does not change ZRS enhancer activity in a reporter assay, but I am not fully convinced this is indeed the case. The HD motif mutant enhancer does look weaker (Fig. 6C), though perhaps the mean intensity from mCH and GFP cannot be directly compared. How would the comparison look if the mutant and wild type ZRS enhancers were swapped in this reporter? The possibility that the HD motif mutations do quantitatively affect ZRS enhancer activity should be at least acknowledged, and it may confound the author's interpretation of the results in Fig. 6D. See also next point, which would be a more direct way to accomplish a separation of function.

We agree with the reviewer that while the overall spatiotemporal expression pattern driven by the ZRS with mutated HD motifs looks similar, the enhancer strength is weakened by approximately 50% in shorter-range assay, albeit not statistically significant (**Fig 6c**). We now acknowledge this result in the revised manuscript and provide a possible explanation:

Results (Pg 17 Ln 20-24): While both the ZRS and ZRS Δ HD directed expression in a spatially highly restricted manner to the ZPA, ZRS Δ HD showed lower, albeit not statistically significant, quantitative activity than the ZRS regardless of the fluorescent reporter pairing. This may be due to the close proximity of [C/T]AATTA HD motifs to the ETS sites, which are critical for ZRS activity⁶⁰.

We also followed the reviewer's suggestion and swapped the reporter genes. We obtained a similar result – ZRS with mutated HD motifs was still active. We added these new results on **Pg 17 Ln 18-20** and as revised **Fig. 6c** and **Extended Data Fig. 7** in the revised manuscript:

Results (Pg 17 Ln 18-20): We obtained a similar result when we swapped fluorescent reporter genes ruling out the influence of fluorophores on our observations (Fig. 6c).

With respect to the separation of long- and short-range functions, work by others has demonstrated that as little as 50% of normal *Shh* expression is sufficient for normal limb outgrowth (Paliou et al. 2019). We, therefore, expect full limb outgrowth for mice with ZRS at 50% strength. However we observe a limbless phenotype indistinguishable from complete *Shh* loss of function in mice with mutated HD motifs in the endogenous ZRS. The remaining difference is likely due to the distance effect. Per the reviewer's suggestion, we also showed that LHX HD motifs are critical for REX function (see the response to next point), which we agree more conclusively demonstrates the separation of long- and short-range function.

Reviewer 3, Specific Point 2: Does mutation of the three HD motif sites within the REX element (Fig. 5D) kill its ability to rescue *Shh* expression and limb formation, when coupled to HS72? This is an important experiment, since in this setup, the extender and enhancer functions can be more easily genetically separated.

We agree with the reviewer that this is a critical experiment. To test this, we generated two additional knock-in mouse lines in which the ZRS was replaced with the extended HS72 sequence containing the REX element with either the single LEF1 motif or both LHX motifs mutagenized. Only knock-in mice with disrupted LHX motifs resulted in mice with truncated limbs, indicating that HD motifs are required for REX function. We added these new results on **Pg 14 Ln 4-12** and as new **Fig. 4** in the revised manuscript:

Results (Pg 14 Ln 4-12): To determine the importance of LHX and LEF1 motifs for REX function, we generated two knock-in mice in which the ZRS was replaced with the extended HS72 sequence containing the REX element with either the single LEF1 motif or both LHX motifs mutagenized. Mice containing the REX element with mutated LEF1 motif showed fully developed limbs with polydactyly indicating that disruption of the LEF1 motif does not abolish long-range activity of the REX element (**Fig. 4b**). However, disrupting both LHX motifs resulted in mice with truncated limbs, indicative of a loss of *Shh* expression in the limb bud (**Fig. 4c**). Together, these results demonstrate that the LHX motifs in the REX element are required for its ability to facilitate long-range enhancer activity at *Shh* locus.

Reviewer 3, Specific Point 3: Beyond the importance of the HD motifs, a current study provides little insight into the mechanism by which extender elements might work. While detailed molecular investigation is – in my view at least – beyond the scope of the current study, a couple of straightforward genomic analyses may provide some additional insights. For example, the [C/T]AATTA motif is bound by many HD motif TFs expressed in the developing limb. Based on the available ChIP-seq data, is there any information on which of these HD TFs may be preferentially binding at the long-range vs short-range hindlimb enhancers (using extended regions, as in Fig. 5)?

We agree that the investigation of upstream factors that regulate REX elements is beyond the scope of this study. We nevertheless followed the reviewers' suggestion and inspected published ChIP-seq and CUT&RUN datasets performed in embryonic limb buds. There are, unfortunately, no publicly available binding datasets for LHX2, LHX9 or LDB1 in developing limb buds.

In order to further investigate this binding, we surveyed an array of mouse hindlimb bud datasets of the binding of limb bud transcription factors with known [C/T]AATTA preferences. This included:

- E11.5 CUT&RUN (ArrayExpress E-MTAB-14340) for DLX5 and LEF1 (the other motif identified at the REX element)
- E11.5 ChIP-seq dataset for PITX1 from GSE41591 (Infante et al. 2013)
- E12.5 ChIP-seq for LXMB1 from GSE84064 (Haro et al. 2017)

In all cases, we see no strong evidence of binding at either the REX element or ZRS. However, the data quality is often not sufficient to derive a definite conclusion.

E11.5 HOXA11 (from GSM3504924) did broadly overlap with the ZRS and the HS72 enhancer and REX element, but calls are too broad to see if they specifically overlap the motifs we disrupted

as opposed to the HOX motifs already known to play an important role in most mesenchymal limb enhancers.

Reviewer 3, Specific Point 4a: Is there an enrichment of cohesin at the long-range enhancers compared to the short-range enhancers?

Per the reviewer's suggestion, we compared the occupancy of Cohesin subunit RAD21 and CTCF (Andrey et al. 2017) in E11.5 hindlimb bud at long- (400 kb - 2 Mb E-P distance) and short-range (10 - 200 kb) VISTA limb enhancers assigned target genes through Hi-C or Multiome (the same set used for motif enrichment). We found no significant differences in occupancy of either factor.

Reviewer 3, Specific Point 4b: Presumably, the REX element is CTCF independent. Can authors confirm that it is not bound by CTCF in the limb?

We agree that the REX element is likely CTCF-independent. We scanned the REX element for the presence of CTCF or YY1 motifs and did not find any. We also inspected a published CTCF ChIP-seq dataset from E11.5 limb buds (Andrey et al. 2017) and determined that neither HS72 nor REX have any CTCF binding *in vivo*. We added these new results on **Pg 13 Ln 22-23** and as new **Extended Data Fig. 6e** in the revised manuscript:

Results (Pg 13 Ln 22-23): The REX element lacks CTCF and YY1 motifs and is not bound by CTCF⁵⁷ in embryonic limb buds (Extended Data Fig. 6e).